# The *Caenorhabditis elegans* protein SAS-5 forms large oligomeric assemblies critical for centriole formation

Kacper B Rogala[1,2], Nicola J Dynes[3], Georgios N Hatzopoulos[1], Jun Yan[4], Sheng Kai Pong[3], Carol V Robinson[4], Charlotte M Deane[2], Pierre Gönczy[3]*, Ioannis Vakonakis[1]*

[1]Department of Biochemistry, University of Oxford, Oxford, United Kingdom; [2]Department of Statistics, University of Oxford, Oxford, United Kingdom; [3]Swiss Institute for Experimental Cancer Research, School of Life Sciences, Swiss Federal Institute of Technology, Lausanne, Switzerland; [4]Department of Chemistry, University of Oxford, Oxford, United Kingdom

**Abstract** Centrioles are microtubule-based organelles crucial for cell division, sensing and motility. In *Caenorhabditis elegans*, the onset of centriole formation requires notably the proteins SAS-5 and SAS-6, which have functional equivalents across eukaryotic evolution. Whereas the molecular architecture of SAS-6 and its role in initiating centriole formation are well understood, the mechanisms by which SAS-5 and its relatives function is unclear. Here, we combine biophysical and structural analysis to uncover the architecture of SAS-5 and examine its functional implications in vivo. Our work reveals that two distinct self-associating domains are necessary to form higher-order oligomers of SAS-5: a trimeric coiled coil and a novel globular dimeric *Implico* domain. Disruption of either domain leads to centriole duplication failure in worm embryos, indicating that large SAS-5 assemblies are necessary for function in vivo.

*For correspondence: pierre. gonczy@epfl.ch (PG); ioannis. vakonakis@bioch.ox.ac.uk (IV)

Competing interests: The authors declare that no competing interests exist.

## Introduction

Most eukaryotes harbor microtubule-based cylindrical organelles called centrioles that exhibit a striking ninefold radial symmetry, and which are crucial for a wide range of cellular functions (reviewed in *Gönczy, 2012*; *Agircan et al., 2014*). In resting cells, centrioles are usually found near the plasma membrane where they organize the formation of flagella and cilia, whereas in proliferating cells centrioles typically reside adjacent to the nucleus, where they recruit pericentriolar material to form the centrosome, the major microtubule organizing center of animal cells. Centrosomes play a major role in directing cellular architecture during interphase and bipolar spindle assembly during mitosis. Centriole numbers are tightly regulated, with centriole duplication occurring only once per cell cycle, in concert with replication of the genetic material (reviewed in *Firat-Karalar and Stearns, 2014*). Abnormalities in centriole formation can impair cell signaling and motility owing to defective cilia or flagella, as well as cause spindle positioning defects and genome instability due to aberrations in centrosome numbers and/or sizes. Thus, it is not surprising that centriolar defects are at the root of multiple medical conditions, including primary microcephaly, male sterility and possibly cancer (reviewed in *Nigg and Raff, 2009*; *Arquint et al., 2014*; *Chavali et al., 2014*; *Godinho and Pellman, 2014*; and *Nachury, 2014*).

Five proteins required for centriole assembly were originally identified in *Caenorhabditis elegans* through genetic analysis and functional genomics (reviewed in *Gönczy, 2012*); these include the recruiting factor SPD-2 (*Kemp et al., 2004*; *Pelletier et al., 2004*), the kinase ZYG-1 (*O'connell et al., 2001*), and the

**eLife digest** Most animal cells contain structures known as centrioles. Typically, a cell that is not dividing contains a pair of centrioles. But when a cell prepares to divide, the centrioles are duplicated. The two pairs of centrioles then organize the scaffolding that shares the genetic material equally between the newly formed cells at cell division.

Centriole assembly is tightly regulated and abnormalities in this process can lead to developmental defects and cancer. Centrioles likely contain several hundred proteins, but only a few of these are strictly needed for centriole assembly. New centrioles usually assemble from a cartwheel-like arrangement of proteins, which includes a protein called SAS-6. In the worm *Caenorhabditis elegans*, SAS-6 associates with another protein called SAS-5. This interaction is essential for centrioles to form, but the reason behind this is not clearly understood.

Now, Rogala et al. have used a range of techniques including X-ray crystallography, biophysics and studies of worm embryos to investigate the role of SAS-5 in *C. elegans*. These experiments revealed that SAS-5 proteins can interact with each other, via two regions of each protein termed a 'coiled-coil' and a previously unrecognized 'Implico domain'. These regions drive the formation of assemblies that contain multiple SAS-5 proteins.

Next, Rogala et al. asked whether SAS-5 assemblies are important for centriole duplication. Mutant worm embryos, in which SAS-5 proteins could not interact with one another, failed to form new centrioles. This resulted in defects with cell division. An independent study by Cottee, Muschalik et al. obtained similar results and found that the fruit fly equivalent of SAS-5, called Ana2, can also self-associate and this activity is required for centriole duplication.

Further work is now needed to understand how SAS-5 and SAS-6 work with each other to form the initial framework at the core of centrioles.

coiled-coil domain containing proteins SAS-5, SAS-6 and SAS-4 (*Kirkham et al., 2003*; *Leidel and Gönczy, 2003*; *Dammermann et al., 2004*; *Delattre et al., 2004*; *Leidel et al., 2005*). Following localization of these five proteins to the site of new centriole formation, recruitment of microtubules completes the assembly process, giving rise to a ninefold- symmetric centriole ~100 nm in diameter (*Pelletier et al., 2006*). Functionally equivalent proteins have now been identified throughout eukaryotes (*Carvalho-Santos et al., 2010*; *Hodges et al., 2010*), indicating an evolutionary shared assembly pathway for centriole formation.

Whereas SAS-6 is critical for establishing the ninefold radial symmetry of centrioles (reviewed in *Gönczy, 2012*; and *Hirono, 2014*), the underlying structural mechanism differs between *C. elegans* and other species. Crystallographic and/or electron microscopic analysis supports the view that recombinant SAS-6 proteins from *Chlamydomonas reinhardtii*, *Danio rerio* and *Leishmania major* form ninefold-symmetric rings (*Kitagawa et al., 2011b*; *Van Breugel et al., 2011*; *Van Breugel et al., 2014*). Such SAS-6 rings are thought to dictate the ninefold- symmetrical assembly of the entire centriole. In contrast, similar analysis of *C. elegans* SAS-6 suggests formation of a spiral oligomer with 4.5-fold symmetry per turn, thus generating ninefold symmetry upon two turns of the spiral (*Hilbert et al., 2013*).

*C. elegans* SAS-6 physically interacts with SAS-5 (*Leidel et al., 2005*; *Qiao et al., 2012*; *Hilbert et al., 2013*; *Lettman et al., 2013*), a protein that shuttles rapidly between the cytoplasm and centrioles throughout the cell cycle (*Delattre et al., 2004*). The presence of SAS-6 and SAS-5 at centrioles is essential for formation of the central tube, a cylindrical structure at the core of the emerging centriole (*Pelletier et al., 2006*). Depletion of SAS-5 (*Dammermann et al., 2004*; *Delattre et al., 2004*) or SAS-5 mutants that are unable to bind SAS-6 (*Delattre et al., 2004*; *Qiao et al., 2012*; *Lettman et al., 2013*) prevent central tube formation, and therefore centriole assembly. Although SAS-5 has been proposed to assist SAS-6 organization (*Qiao et al., 2012*; *Lettman et al., 2013*), the mechanisms by which this may be achieved are not known, in part because the architecture of SAS-5 has not yet been resolved.

Here, we employ biophysical methods and X-ray crystallography, together with functional assays in *C. elegans* embryos, to demonstrate that large assemblies of SAS-5 are necessary for centriole formation. Our results lead us to propose a working model in which SAS-5 oligomers may assist function by providing a multivalent framework for the assembly of *C. elegans* SAS-6 oligomers.

## Results

### SAS-5 features two independently folded domains

Previous attempts at recombinant expression of full-length *C. elegans* SAS-5 (SAS-5$_{FL}$, amino acids 1–404, *Figure 1A*) yielded insoluble or marginally soluble material (*Qiao et al., 2012*; *Lettman et al., 2013*). To tackle this problem, we constructed a bacterial expression vector system harboring 13 different solubility-tags, which allowed us to obtain soluble SAS-5$_{FL}$ fused to MsyB (*Zou et al., 2008*) in quantities sufficient for biophysical analysis (*Figure 1—figure supplement 1*, which shows all recombinant proteins in this study). Characterization of such purified SAS-5$_{FL}$ using circular dichroism (CD) revealed the presence of protein aggregates (*Figure 1—figure supplement 2A,B*), a conclusion also supported by size-exclusion chromatography multi-angle light scattering (SEC-MALS, *Figure 1—figure supplement 2C*) and negative-stain electron microscopy (*Figure 1—figure supplement 2D*). We sought to locate the motif in SAS-5 that causes aggregation by expressing a series of MsyB-tagged C-terminal truncation constructs, and observed a significant increase in solubility when comparing a construct encompassing residues 2–279 with one containing residues 2–296 (*Figure 1—figure supplement 3*). Secondary structure prediction suggested a β-strand between residues 282 and 295 (*Figure 1—figure supplement 4*). Excising this region from untagged SAS-5$_{FL}$ (SAS-5$_{Δ282–295}$) or replacing it with a Gly-Ser-Ala-rich flexible linker of equal length (SAS-5$_{FLEX}$) resulted in proteins with CD spectra characteristic of mainly α-helical proteins (*Figure 1B*). These results suggest that the predicted β-strand between residues 282–295 of SAS-5 drives formation of large protein aggregates.

Interestingly, thermal unfolding of the soluble SAS-5$_{Δ282–295}$ and SAS-5$_{FLEX}$ variants showed a two-step cooperative melting process, thus revealing the presence of two independently folded domains (*Figure 1C*). Sequence-based prediction suggested the presence of a coiled coil spanning residues 125–180, as well as of three tightly-spaced α-helices (residues 210–265); these two elements are separated from each other by a presumed disordered linker of ∼30 residues (*Figure 1—figure supplement 4*). We expressed both the coiled coil (SAS-5$_{CC}$) and the predicted helical region (SAS-5$_{Imp}$ for *Implico*, see below) and performed CD analysis. SAS-5$_{CC}$ displayed a double-minimum spectrum (*Figure 1D*) characteristic of α-helical coiled coils, and exhibited moderate thermal stability in isolation (apparent melting transition temperature, T$_m$ ∼ 37°C, *Figure 1E*). In contrast, CD of SAS-5$_{Imp}$ revealed a very stable (T$_m$ ∼ 72°C) α-helical domain (*Figure 1D,E*). A SAS-5 construct that combined both the coiled coil and the α-helical domain (SAS-5$_{125–265}$, residues 125–265) showed CD spectra and a two-step thermal unfolding profile highly similar to that of SAS-5$_{Δ282–295}$ and SAS-5$_{FLEX}$ (compare *Figure 1F,G* with *Figure 1B,C*). We attribute the first melting transition of SAS-5$_{125–265}$ (T$_m$ of ∼50°C) to unfolding of the coiled-coil domain and the second (T$_m$ of ∼70°C) to unfolding of the *Implico* domain. Moreover, CD spectra of isolated SAS-5 N-terminal (residues 2–122) or C-terminal (269–404) fragments showed no persistent secondary structure or cooperative thermal unfolding (*Figure 1—figure supplement 5*), consistent with similar previous analysis of the SAS-5 N-terminus (*Shimanovskaya et al., 2013*). Overall, these findings led us to conclude that the segment between residues 125–265 contains two independently folded domains and encompasses all structured elements of SAS-5.

### The SAS-5 coiled coil forms a moderately stable trimer

We next set out to determine the X-ray crystallographic structures of the two independently folded domains. Native crystals of SAS-5$_{CC}$ diffracted to 1.8 Å resolution, whilst phases for structure solution were determined by single anomalous dispersion (SAD) using trimethyl lead acetate derivatized crystals (*Tables 1, 2*). The structure of SAS-5$_{CC}$ suggested a parallel trimeric coiled coil, although this arrangement is distorted by crystal packing that intercalates the ends of successive triple helical bundles (*Figure 2A*). To correct for these distortions, we performed molecular dynamics (MD) simulations starting from the crystallographic structure of SAS-5$_{CC}$, which rapidly converged to a trimeric coiled-coil structure (*Figure 2B–D*). SEC-MALS analysis of a slightly longer coiled-coil construct (SAS-5$_{CC-L}$, residues 101–206) demonstrated formation of a trimer in solution in a concentration-dependent manner, with apparent K$_d$ of ∼3 µM (*Figure 2G,H*). Both the crystallographic model and MD simulations show a series of hydrophobic residues (W133, M137, L141, I144, I148, L159, M167 and M171) forming the core of the coiled coil and enabling trimer formation (*Figure 2E,F*). Interestingly, we found that all these residues are conserved or conservatively

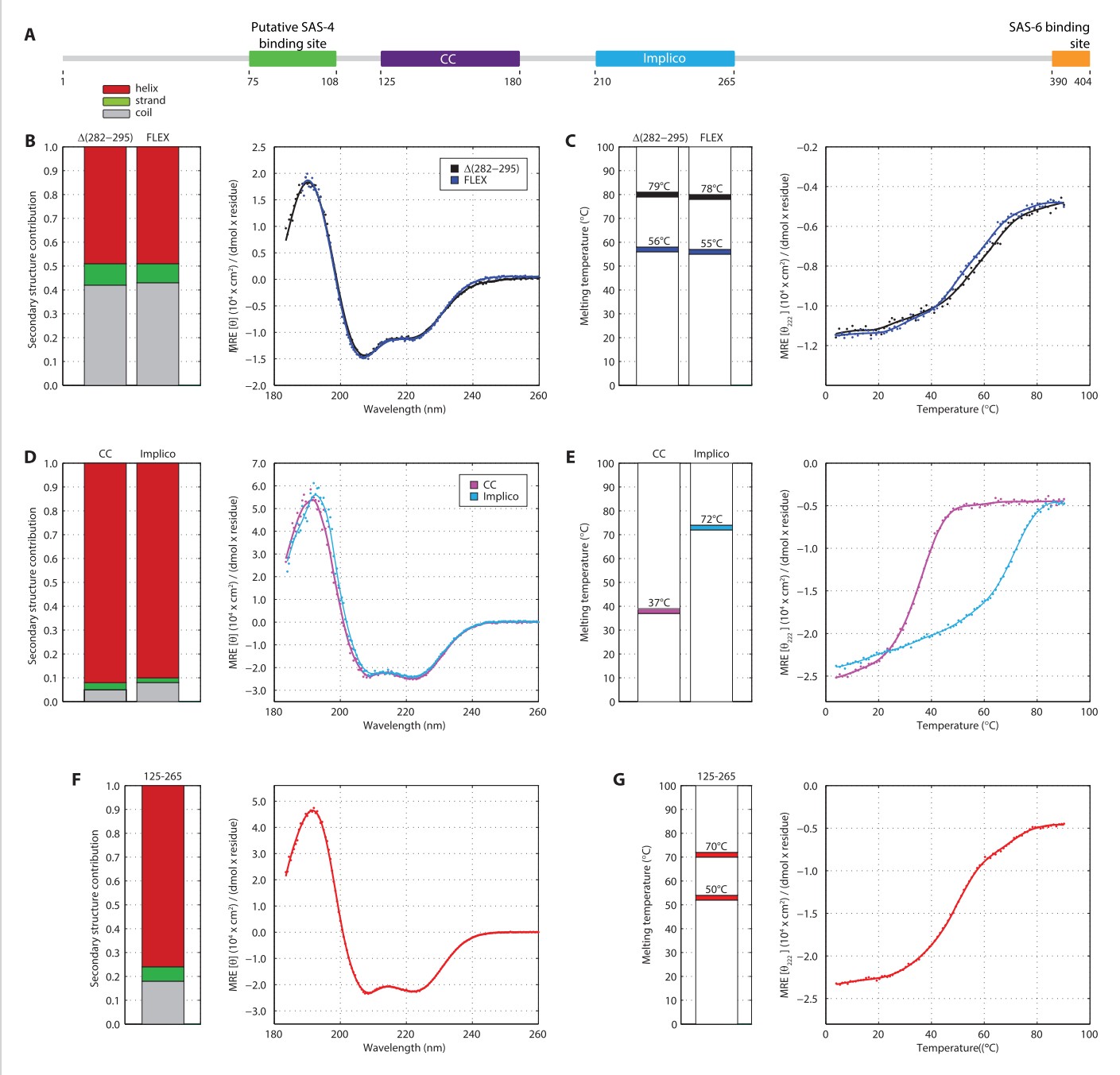

**Figure 1**. SAS-5 comprises two independently folded domains. (**A**) Schematic representation of SAS-5 architecture showing the relative locations and residue boundaries of the coiled-coil and *Implico* domains, the putative SAS-4 binding site (**Hatzopoulos et al., 2013**) and the SAS-6 binding site (**Qiao et al., 2012**; **Hilbert et al., 2013**). (**B**) Overlaid CD spectra of SAS-5$_{\Delta282-295}$ and SAS-5$_{FLEX}$ samples recorded at 10°C, shown as per residue molar elipticity vs wavelength. The semi-quantitative contribution of secondary structure elements in each spectrum is deconvoluted in the bar charts on the left. Grey color corresponds to random coil, green to β-strand and red to α-helical segments. (**C**) Thermal unfolding profiles of the same samples monitored by recording molar elipticity at 222 nm as a function of temperature, and graphical representation of the melting transition temperatures observed in each sample. (**D**–**G**) Similar CD spectra and thermal unfolding profiles of (**D**,**E**) SAS-5$_{CC}$ and SAS-5$_{Imp}$, and (**F**,**G**) SAS-5$_{125-265}$ samples.

The following figure supplements are available for figure 1:

**Figure supplement 1**. Recombinant proteins sample quality.

**Figure supplement 2**. SAS-5 forms protein aggregates in vitro.
*Figure 1. continued on next page*

*Figure 1. Continued*

**Figure supplement 3**. A short SAS-5 segment promotes protein aggregation.

**Figure supplement 4**. SAS-5 secondary structure and disorder predictions.

**Figure supplement 5**. The SAS-5 N- and C-terminal segments are unstructured in isolation.

substituted in SAS-5 homologs in other *Caenorhabditis* species, suggesting functional relevance (*Figure 2—figure supplement 1*). To investigate the stability of the coiled-coil domain across nematode evolution, we expressed constructs corresponding to SAS-5$_{CC-L}$ from *Caenorhabditis brenneri*, *Caenorhabditis briggsae*, *Caenorhabditis remanei*, *Caenorhabditis sinica* and *Caenorhabditis tropicalis*. In all cases, SEC-MALS analysis showed similar oligomerization properties and apparent affinities as for *C. elegans* SAS-5$_{CC-L}$ (*Figure 2—figure supplement 2*). Taken together, these data suggest that the trimeric coiled coil is an evolutionary conserved feature of nematode SAS-5 proteins.

## SAS-5$_{Imp}$ is a structurally novel dimeric domain

We next turned our attention to the second independently folded domain, SAS-5$_{Imp}$. We found that this newly recognized domain is highly conserved amongst *Caenorhabditis* species (*Figure 3—figure supplement 1*). Importantly, SEC-MALS experiments revealed that SAS-5$_{Imp}$ forms a stable dimer (*Figure 3A*). Crystals of SAS-5$_{Imp}$ diffracted to 1.0 Å resolution and experimental phases were determined by SAD from mercury acetate derivatized samples (*Table 1*). The SAS-5$_{Imp}$ structure revealed a dimer composed of interlocked chains arranged in an antiparallel fashion (*Figure 3B*). Each chain features three α-helices (A1 to A3 and B1 to B3) that interact in a pairwise manner (A1–B3, A2–B2 and A3–B1). Proline residues at the linkers between helices allow for tight 90° turns (*Figure 3C*), resulting in a very compact dimer of just ~4 nm in diameter. The dimer interface (*Figure 3D*) involves the majority of hydrophobic core residues (I219, I225, A229, L230, I232, I233, L237, F243, I247, V250, L251); thus, breaking the dimer is expected to unfold the SAS-5$_{Imp}$ domain. Given the high thermal stability of SAS-5$_{Imp}$ (*Figure 1E*), this suggests formation of stable dimers even at very low protein concentrations. This interlocked tight dimer is a novel structural feature that we named *Implico*, from Latin *to entangle*.

## The molecular architecture of SAS-5 drives the formation of higher-order assemblies

Together, the two oligomerization domains of SAS-5 have the potential to drive formation of protein assemblies of higher-order than merely dimers or trimers. We envision that SAS-5 would readily dimerize even at low concentrations via the *Implico* domain, as oligomerization of this domain is tighter compared to that of the SAS-5 coiled coil. Such SAS-5 dimers could then come together to form higher-order configurations. To test this possibility, we performed SEC-MALS experiments on MsyB-tagged SAS-5$_{FLEX}$, which revealed concentration-dependent oligomerization of SAS-5 from dimers to an approximately equimolar population of tetramers and hexamers (*Figure 4A,B*). Analogous results were obtained with a MsyB-tagged SAS-5 construct that lacks the disordered protein C-terminus (SAS-5$_{2–265}$) (*Figure 4C,D*). Moreover, native mass-spectrometry of SAS-5$_{2–265}$ also supported the presence of higher-order protein assemblies up to hexamers in solution (*Figure 4E*). Taken together, these data support the notion that the SAS-5 coiled coil promotes the association of protein dimers towards higher-order assemblies. Furthermore, our results are consistent with an earlier SEC-MALS analysis of MBP-tagged SAS-5 constructs of residues 1–260, which showed formation of tetramers at an unspecified protein concentration (*Shimanovskaya et al., 2013*).

To further probe the architectural model of higher-order SAS-5 assemblies, we proceeded to disrupt each oligomerization interface by engineering charged residues in place of core hydrophobic amino acids. The CD spectra of the resulting SAS-5$_{CC}$ L141E mutant showed significant reduction of α-helical content in the coiled coil and a striking reduction in stability ($T_m < 10°C$, *Figure 5—figure supplement 1A,B*), whereas that of the SAS-5$_{Imp}$ I247E mutant showed absence of α-helical structure and complete loss of cooperativity upon thermal unfolding (*Figure 5—figure supplement 1C,D*).

**Table 1**. Crystallographic data collection and refinement statistics

| Protein | SAS-5$_{CC}$ | SAS-5$_{CC}$ (Pb-derivative) | SAS-5$_{Imp}$ | SAS-5$_{Imp}$ (Hg-derivative) |
|---|---|---|---|---|
| PDB code | 4YV4 | – | 4YNH | – |
| Space group | P 2$_1$ 2$_1$ 2$_1$ | P 2$_1$ 2$_1$ 2$_1$ | P 1 2$_1$ 1 | P 1 2$_1$ 1 |
| Unit cell (Å, °) | a = 46.52; | a = 46.32; | a = 27.18; | a = 27.85; |
| | b = 55.10; | b = 52.99; | b = 36.29; | b = 36.65; |
| | c = 191.50 | c = 191.55 | c = 42.60; | c = 42.13; |
| | | | β = 97.50 | β = 97.74 |
| Beamline | DLS/I04 | DLS/I03 | ESRF ID14-4 | DLS/I02 |
| Wavelength (Å) | 0.9795 | 0.947 | 0.900 | 1.007 |
| Resolution range (Å) | 41.90-1.80 | 41.70-1.90 | 18.49-1.00 | 41.75-1.70 |
| High resolution shell (Å) | 1.84-1.80* | 1.95-1.90† | 1.05-1.00 | 1.74-1.70 |
| R$_{merge}$‡ | 0.047 (1.092) | 0.089 (1.088) | 0.108 (0.763) | 0.065 (0.541) |
| R$_{pim}$‡ | 0.034 (0.988) | 0.014 (0.680) | 0.044 (0.314) | 0.017 (0.151) |
| Completeness‡ (%) | 98.1 (97.8) | 80.1 (33.6)§ | 99.4 (98.3) | 85.6 (39.4)# |
| Multiplicity‡ | 2.9 (2.9) | 9.3 (3.6) | 6.9 (6.8) | 18.4 (13.6) |
| Mean I/σ(I)‡ | 7.4 (0.7) | 13 (0.9) | 12.0 (4.2) | 26.5 (4.5) |
| Phasing | | | | |
| No. of heavy atom sites | – | 9 | – | 2 |
| Resolution | – | 41.70-1.90 | – | 41.75-1.70 |
| FOM *initial*¶ | – | 0.39 | – | 0.39 |
| FOM *DM*** | – | 0.61 | – | 0.75 |
| Refinement statistics | | | | |
| R$_{work}$ (reflections) | 20.9% (30,118) | – | 12.8% (42,016) | – |
| R$_{free}$(reflections) | 23.8% (1600) | – | 14.9% (2232) | – |
| Number of atoms | | | | |
| Protein atoms | 3647 | – | 2040 (including H) | – |
| Ligands | 25 | – | – | – |
| Water | 261 | – | 124 | – |
| Average B factors (Å$^2$) | | | | |
| Protein atoms | 50.9 | – | 10.9 | – |
| Water | 46.5 | – | 23.5 | – |
| RMSD from ideal values | | | | |
| Bonds / angles (Å/°) | 0.01 / 0.95 | – | 0.007 / 1.152 | – |
| MolProbity statistics†† | | | | |
| Ramachandran favored (%) | 99.5% | – | 99.2% | – |
| Ramachandran disallowed (%) | 0% | – | 0% | – |
| Clashscore (percentile) | 2.49 (99$^{th}$) | – | 0.98 (98$^{th}$) | – |
| MolProbity score (percentile) | 1.11 (100$^{th}$) | – | 0.79 (99$^{th}$) | – |

*Anisotropic diffraction of 2.17 Å, 2.59 Å and 1.73 Å highest resolution along the a*, b* and c* axes, respectively, based on a mean I/σ(I) > 2.0 criterion.
†Anisotropic diffraction of 2.18 Å, 3.03 Å and 2.17 Å highest resolution along the a*, b* and c* axes, respectively, based on a mean I/σ(I) > 2.0 criterion.
‡By Aimless (**Evans and Murshudov, 2013**), values in parentheses correspond to the high resolution shell.
§98.0% complete to 2.26 Å
#98.8% complete to 2.03 Å
¶From PHASER (**Mccoy et al., 2007**).
**From RESOLVE (**Terwilliger, 2000**).
††From MolProbity (**Chen et al., 2010**).

**Table 2**. Anisotropy correction statistics* for native SAS-5$_{CC}$ crystallographic data

| Resolution (Å) | Number of observed reflections | | Redundancy | | Completeness (%) | | R$_{merge}$ (%) | | Mean I/σ(I) | |
|---|---|---|---|---|---|---|---|---|---|---|
| | Before | After | Before | After | Before | After | Before | After | Before | After |
| 8.05 | 1428 | 1428 | 2.3 | 2.3 | 92.3 | 92.3 | 2.9 | 2.9 | 22.3 | 21.9 |
| 5.69 | 2617 | 2598 | 2.5 | 2.5 | 92.1 | 92.2 | 3.3 | 3.3 | 22.3 | 21.9 |
| 4.65 | 3322 | 3317 | 2.6 | 2.6 | 94.4 | 94.4 | 4.0 | 4.0 | 22.4 | 22 |
| 4.03 | 3949 | 3923 | 2.6 | 2.6 | 93.3 | 93.2 | 3.6 | 3.6 | 22.7 | 22.3 |
| 3.6 | 4581 | 4527 | 2.7 | 2.7 | 95.8 | 95.8 | 3.7 | 3.8 | 21.7 | 21.3 |
| 3.29 | 4925 | 4957 | 2.6 | 2.6 | 95.1 | 95.3 | 4.1 | 4.1 | 19.8 | 19.4 |
| 3.04 | 5760 | 5703 | 2.8 | 2.8 | 96.9 | 96.7 | 5.0 | 5.0 | 17.1 | 16.8 |
| 2.85 | 6026 | 6003 | 2.8 | 2.8 | 96.3 | 96.4 | 5.1 | 5.1 | 14.2 | 13.9 |
| 2.68 | 6696 | 6649 | 2.9 | 2.9 | 97.8 | 97.8 | 6.5 | 6.4 | 12.1 | 11.9 |
| 2.55 | 7098 | 7098 | 2.9 | 3 | 98.6 | 98.5 | 7.5 | 7.5 | 10.1 | 9.9 |
| 2.43 | 7178 | 7136 | 2.8 | 2.8 | 98.2 | 98.1 | 9.4 | 9.3 | 8.2 | 8.1 |
| 2.32 | 7824 | 7781 | 3 | 3 | 98.9 | 99.1 | 11.8 | 11.7 | 6.8 | 6.7 |
| 2.23 | 7844 | 7233 | 2.8 | 2.6 | 97.4 | 91.1 | 15.9 | 14.9 | 5.2 | 5.4 |
| 2.15 | 8551 | 6609 | 3 | 2.3 | 99.0 | 79.6 | 20.4 | 15.9 | 4.4 | 5.3 |
| 2.08 | 8757 | 5726 | 3 | 2 | 99.0 | 66.9 | 29.3 | 20.0 | 3.1 | 4.4 |
| 2.01 | 8707 | 4542 | 2.9 | 1.5 | 98.7 | 51.8 | 41.8 | 22.9 | 2.3 | 4.1 |
| 1.95 | 9329 | 3355 | 3 | 1.1 | 99.1 | 33.9 | 59.1 | 24.0 | 1.7 | 4.2 |
| 1.9 | 9150 | 2341 | 2.9 | 0.7 | 98.1 | 20.5 | 93.3 | 30.2 | 1.1 | 3.8 |
| 1.85 | 9824 | 1532 | 3 | 0.5 | 99.2 | 11.6 | 114.0 | 28.6 | 0.9 | 4.5 |
| 1.8 | 9748 | 502 | 2.8 | 0.1 | 97.9 | 3.9 | 143.1 | 31.8 | 0.8 | 3.9 |
| total | 133314 | 92,960 | 2.9 | 2 | 97.6 | 68.2 | 4.9 | 4.4 | 8.2 | 11.3 |

*Derived from the UCLA Diffraction Anisotropy Server (**Strong et al., 2006**).

We next engineered the same substitutions in the SAS-5$_{2–265}$ construct to selectively disrupt each oligomerization interface whilst leaving the other intact. We found that the thermal unfolding profile of a SAS-5$_{2–265}$ L141E mutant showed a reduction in stability of the coiled coil by approximately 25°C, whereas stability of the *Implico* domain was not affected (*Figure 5A,B*). Substituting an additional core hydrophobic residue of the coiled coil (SAS-5$_{2–265}$ L141E/M167E) effectively abrogated coiled coil formation, again with no effect on the *Implico* domain (*Figure 5A,B*). As anticipated, coiled coil disruptions in either SAS-5$_{2–265}$ L141E or L141E/M167E mutants yielded a stable dimer in SEC-MALS experiments (*Figure 5C,D*), mediated by the *Implico* domain. Furthermore, we found that a SAS-5$_{2–265}$ I247E mutant, where the *Implico* domain is disrupted, displayed a single-step thermal unfolding profile as measured by CD with a T$_m$ similar to that of the SAS-5 coiled coil in isolation (compare *Figure 5A,B* to *Figure 1E*). SEC-MALS analysis of the same construct shows a concentration-dependent trimerization, again consistent with formation of just the SAS-5 coiled coil (*Figure 5E,F*). Finally, a SAS-5 mutant where both oligomerization interfaces are disrupted, SAS-5$_{2–265}$ L141E/I247E, showed no cooperativity upon thermal unfolding (*Figure 5A,B*) and remained monomeric in solution (*Figure 5G*). Together, our results support the notion that the two SAS-5 oligomerization domains self-associate independently from one another, and that together they allow formation of higher-order SAS-5 assemblies.

## Higher-order assemblies of SAS-5 are required for centriole duplication in vivo

We next sought to assess the role of higher-order SAS-5 assemblies in vivo. To this end, we generated *C. elegans* transgenic animals expressing either GFP-SAS-5 (wild-type), or GFP-SAS-5 L141E or GFP-SAS-5 I247E mutant versions. All exons of the *sas-5* sequence were recoded in the transgenic

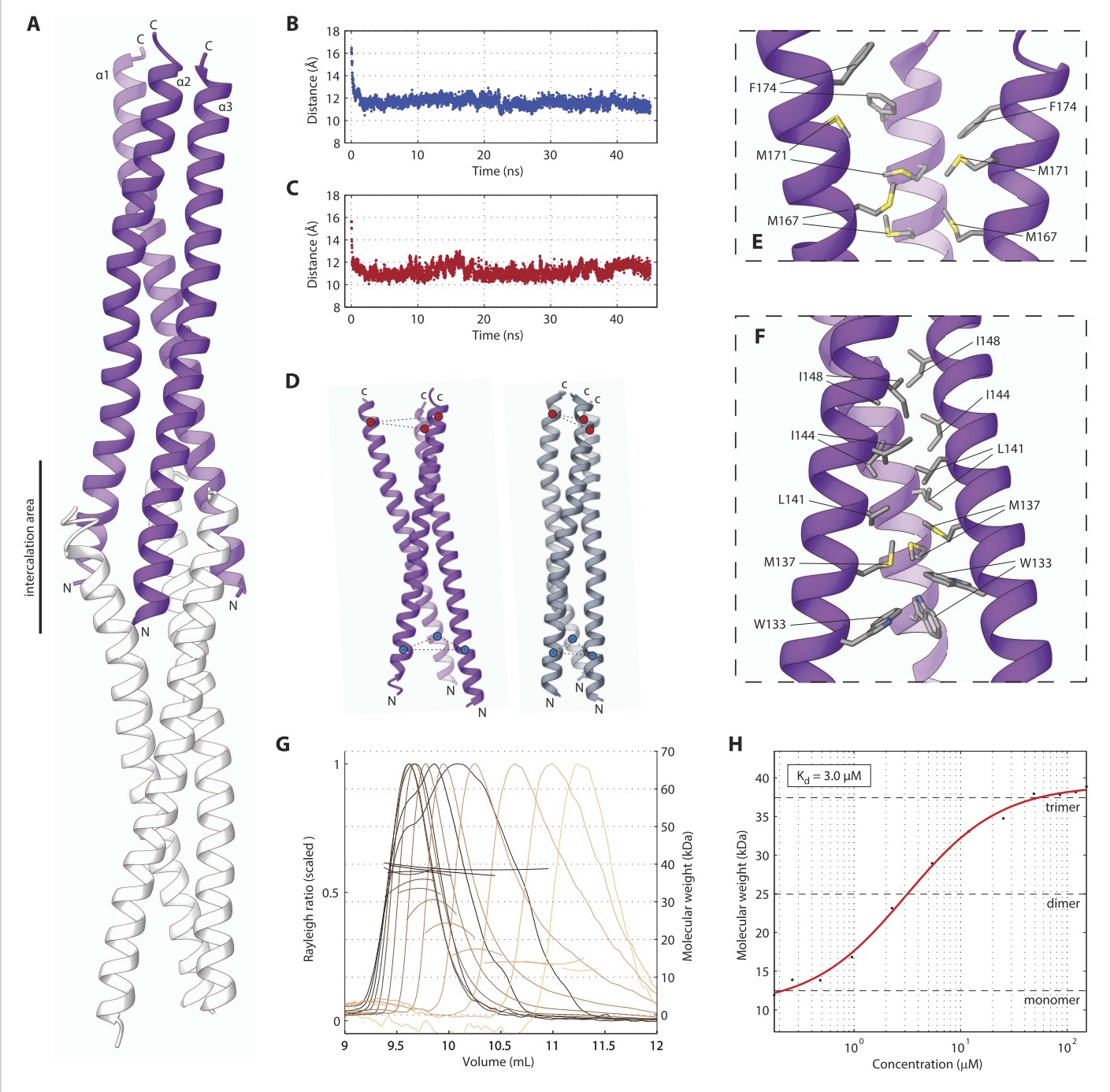

**Figure 2**. The SAS-5 coiled coil forms a parallel trimer. (**A**) Schematic representation of the SAS-5$_{CC}$ crystallographic structure in purple. The protein N- and C-termini are indicated, as is the area where successive molecules intercalate in the crystal, leading to distortions of the coiled-coil structure; an intercalating molecule is shown here in white. (**B**,**C**) Average distance between the centers of the three α-helices at the N-terminus (**B**) or the C-terminus (**C**) of the coiled coil as a function of time in MD simulations. The distances at the initially frayed ends of the coiled coil decreased rapidly from ~16 nm to ~11 nm as the structure converged to a canonical trimeric arrangement. (**D**) Schematic representation of the coiled-coil structure at the start (left, purple) and the end (right, grey) of MD simulations. The helix centers whose distances are plotted in panels **B** and **C** are shown as blue and red spheres, respectively. (**E**,**F**) Magnified view of the coiled-coil hydrophobic core from the crystallographic structure. Residues at the C-terminus (**E**) and N-terminus (**F**) of the coiled coil are shown as sticks; charged substitutions at L141 and M167 disrupted coiled-coil oligomerization. (**G**) Overlay of SEC-MALS chromatograms of SAS-5$_{CC-L}$ at multiple concentrations, showing scaled light scattering intensity vs elution volume. The calculated molecular weight for each trace is shown as continuous line over the chromatogram peak, and corresponds to the right–hand axis. Chromatograms of samples at the highest two concentrations show evidence of protein overloading on the size-exclusion column. (**H**) Plot of average molecular weight

*Figure 2. continued on next page*

*Figure 2. Continued*
from SEC-MALS analysis as a function of on-column protein concentration. The apparent $K_d$, and molecular sizes of monomers, dimers and trimers are indicated.
The following figure supplements are available for figure 2:

**Figure supplement 1**. Alignment of SAS-5 coiled coils from the *Caenorhabditis* genus.

**Figure supplement 2**. The trimeric SAS-5 coiled coil is an evolutionary conserved feature.

constructs to confer resistance to RNAi directed against endogenous *sas-5*. Worms were subjected to *sas-5* RNAi and the resulting embryos analyzed by time-lapse differential interference contrast (DIC) microscopy as a means to assay centriole formation. In wild-type embryos (*Figure 6A* and *Video 1*), the sperm contributes the sole pair of centrioles to the newly fertilized embryo. Following the first

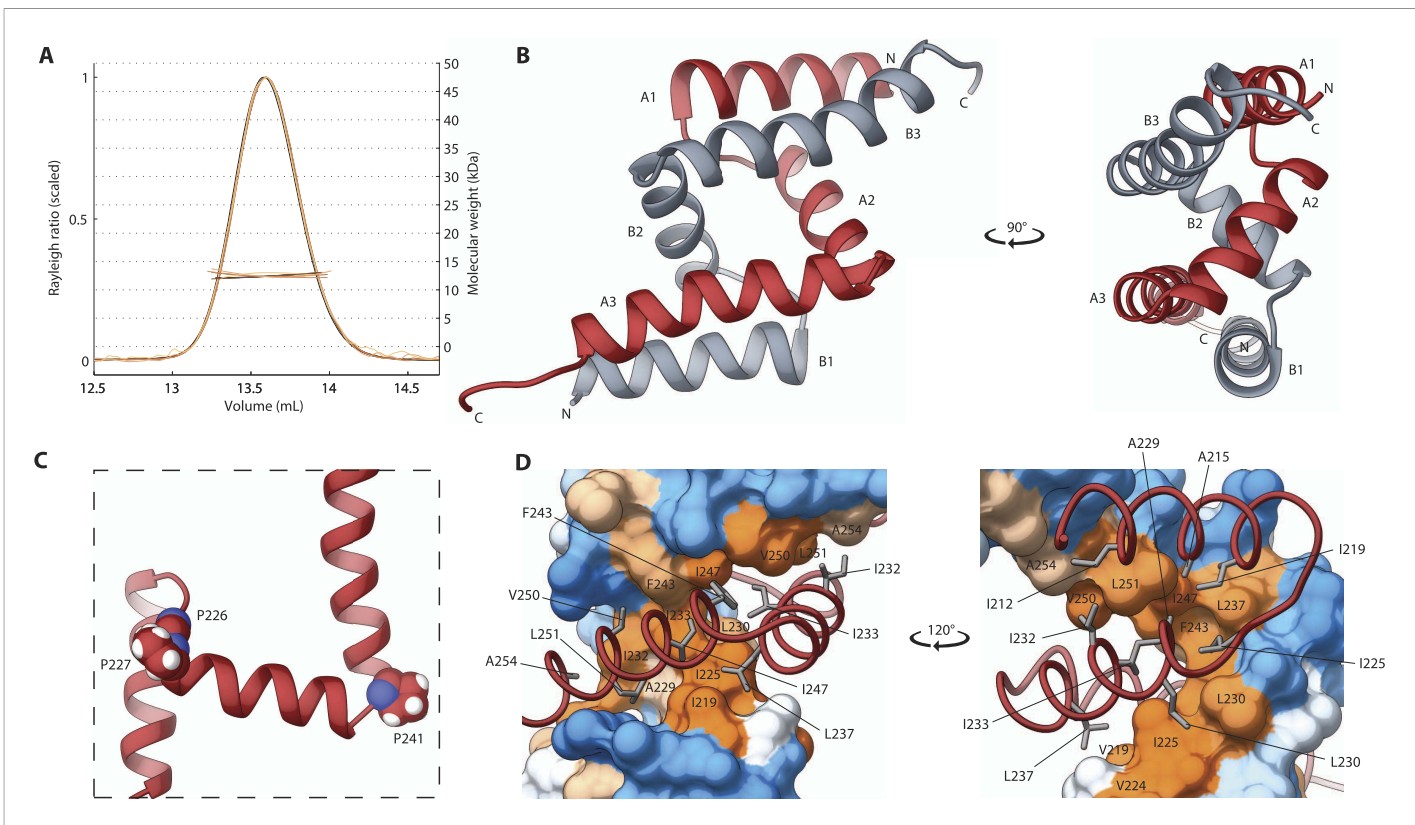

**Figure 3**. The SAS-5 Implico domain comprises a novel type of protein dimer. (**A**) Overlay of SEC-MALS chromatograms of SAS-5_Imp in multiple concentrations showing formation of a stable dimer. (**B**) Two orthogonal views of the SAS-5_Imp crystallographic structure in schematic representation. The α-helices are denoted A1–A3 and B1–B3 for the two protein chains (red and grey, respectively). The N- and C-termini are indicated. (**C**) Magnified view of a single SAS-5_Imp protein chain in schematic representation, showing the proline residues (space-filling representation) at the tight turns between α-helices. (**D**) Two views of the SAS-5_Imp dimeric interface, with one chain represented as surface and the other as ribbon. Hydrophobic residues at the dimer interface are shown as sticks on the ribbon chain, and their location is indicated on the surface. Surface colors are a gradient from blue to orange representing residue hydrophobicity in the Kyte-Doolittle scale (**Kyte and Doolittle, 1982**). Blue corresponds to −4.5 in this scale (most hydrophilic), white to 0 and orange to 4.5 (most hydrophobic).
The following figure supplement is available for figure 3:

**Figure supplement 1**. Alignment of SAS-5 Implico domains from the *Caenorhabditis* genus.

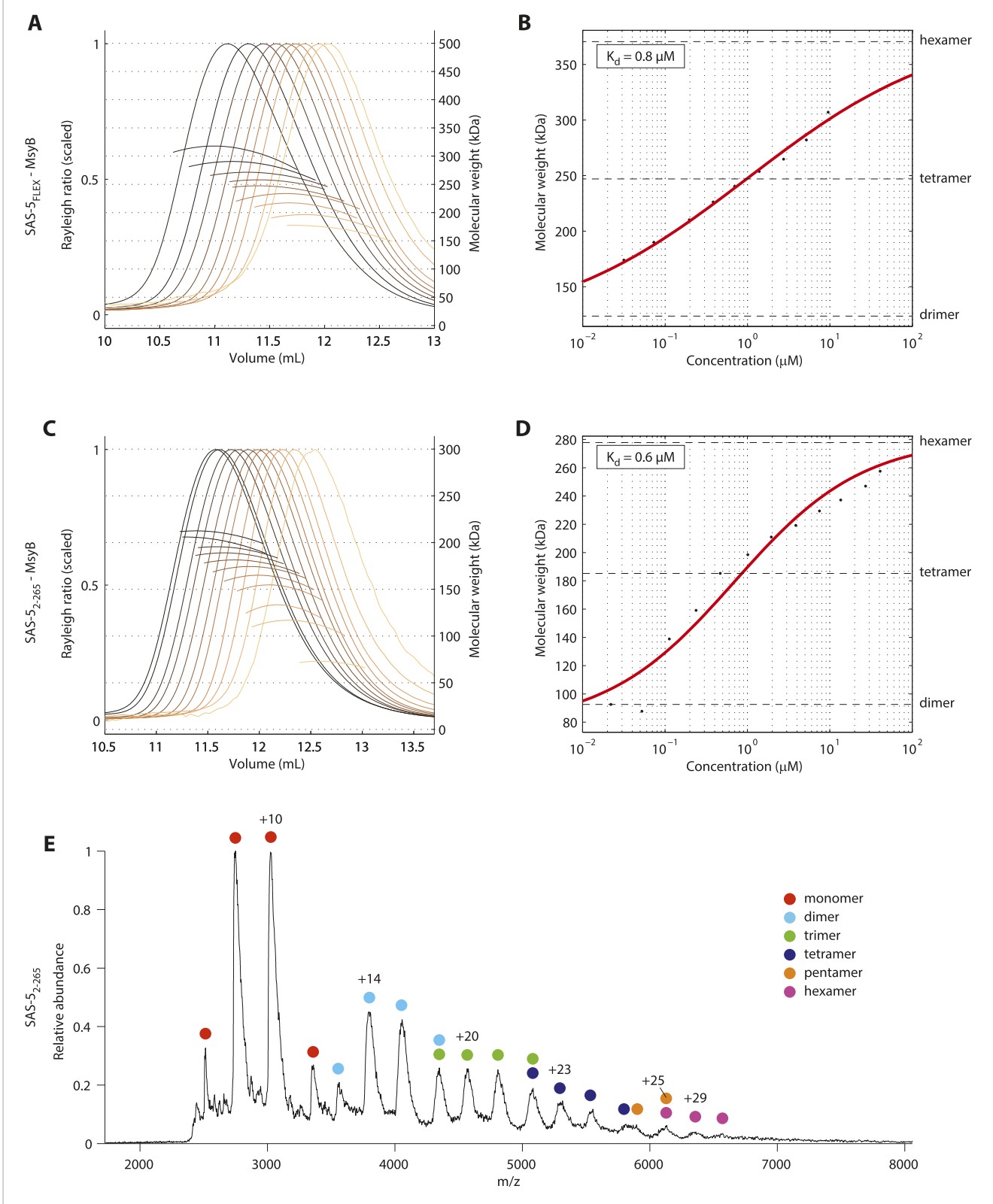

**Figure 4**. SAS-5 forms higher-order assemblies in solution. (**A**) Overlay of SEC-MALS chromatograms of MsyB-tagged SAS-5$_{FLEX}$ in multiple concentrations and (**B**) plot of average molecular weight from SEC-MALS analysis as a function of on-column protein concentration. The apparent $K_d$ and molecular sizes of dimers, tetramers and hexamers are indicated. SAS-5$_{FLEX}$-MsyB reaches an equimolar hexamer to tetramer ratio at the highest

*Figure 4. continued on next page*

*Figure 4. Continued*

concentration we could assess. Fitting the experimental data under the assumption of ultimate hexamer formation yielded $K_d$ values comparable to those of the SAS-5 coiled coil in isolation (compared B with *Figure 2H*). (**C,D**) Similar SEC-MALS analysis of MsyB-tagged SAS-5$_{2–265}$. SAS-5$_{2–265}$-MsyB reaches a 3:1 hexamer to tetramer ratio at the highest concentration point. (**E**) Native mass-spectrometry electrospray ionization spectrum of a 20 µM sample of SAS-5$_{2–265}$, showing relative abundance of protein species as a function of mass to charge ratio. The charged states and protein oligomeric forms corresponding to specific peaks are indicated. Odd-numbered oligomeric forms (monomers, trimers etc) likely correspond to in-flight breakdown of higher- order protein assemblies.

round of centriole duplication in the zygote, the two centrosomes, now each with a pair of centrioles, direct assembly of a bipolar spindle during the first mitotic division. At the two-cell stage, centrioles separate and duplicate, leading to bipolar spindle assembly in each blastomere and resulting in a signature 4-cell stage configuration following mitotic exit (*Figure 6A* and *Video 1*). In *sas-5(RNAi)* embryos, the sperm, which is not affected by RNAi under these experimental conditions, contributes two normal centrioles to the zygote; these paternally contributed centrioles each recruit pericentriolar material and direct bipolar spindle assembly during the first mitotic division (*Delattre et al., 2004*). However, due to depletion of the maternal pool of SAS-5, centriole duplication does not occur, leading to monopolar spindle assembly in each blastomere at the second cell cycle (*Figure 6B*, *Video 2*). As a result, a four-cell configuration is never observed in *sas-5(RNAi)* embryos (*Figure 6B*).

We confirmed the expression of the GFP fusion proteins by Western blot analysis and live imaging after *sas-5* RNAi (*Figure 6—figure supplement 1B–D,G*). Importantly, using time-lapse DIC microscopy to analyze the three transgenic lines generated in this study, we found that expression of GFP-SAS-5 fully rescued the *sas-5(RNAi)* phenotype, allowing all analyzed embryos to progress to the 4-cell stage (*Figure 6C* and *Video 3*), as in untreated wild-type embryos. In stark contrast, GFP-SAS-5 mutants that disrupt either the coiled coil (L141E) or the *Implico* oligomerization domain (I247E) were unable to rescue the *sas-5(RNAi)* phenotype, such that all embryos underwent monopolar spindle assembly in each blastomere at the second cell cycle and thus failed to result in a 4-cell configuration (*Figure 6D,E*, and *Videos 4, 5*). We conclude that both SAS-5 oligomerization interfaces, and hence formation of SAS-5 assemblies larger than merely dimers or trimers, are essential for SAS-5 function in vivo.

We next investigated the distribution of the fusion proteins in one-cell stage embryos treated with *sas-5* (RNAi) by live imaging. Whereas GFP-SAS-5 was observed in foci at centrosomes (*Figure 6—figure supplement 1B*), foci were not seen for either of the oligomerization mutant versions (*Figure 6—figure supplement 1C,D*). In order to investigate the localization of the fusion proteins using a more sensitive approach, we analyzed fixed one-cell stage embryos by immunofluorescence with antibodies against GFP and the centriolar marker IFA (*Figure 6F–H*). We found that whereas GFP-SAS-5 is enriched at centrioles (*Figure 6F,I* and *Delattre et al., 2004*), as was also observed by live imaging, the situation differed for both oligomerization interface mutants. In the case of GFP-SAS-5 L141E, only ~30% of centrosomes exhibited a clear GFP focus, with the remaining centrosomes exhibiting either no GFP focus, or only a weak focus (*Figure 6G,I*). In the case of GFP-SAS-5 I247E, the vast majority of embryos exhibited no detectable GFP focus at centrioles (*Figure 6H,I*). We conclude that SAS-5 oligomerization is important for centriole localization, with the *Implico* domain playing a particularly critical role in this respect. Moreover, the presence of GFP-SAS-5 L141E at centrioles in one third of embryos, combined with the fact that this mutant completely failed to rescue the phenotype, raises the possibility that formation of higher-order SAS-5 assemblies is essential for function even if centriolar localization has been achieved.

## Discussion

*C. elegans* SAS-5 lies at the heart of centriole duplication, given that it associates with SAS-6 (*Leidel et al., 2005*; *Qiao et al., 2012*; *Hilbert et al., 2013*; *Lettman et al., 2013*), a critical structural component for the establishment of the ninefold symmetry of centrioles (*Kitagawa et al., 2011b*; *Van Breugel et al., 2011*; *Hilbert et al., 2013*; *Van Breugel et al., 2014*), and it is required for SAS-6 centriolar localization. Previous biophysical analysis of the SAS-5/SAS-6 interaction was performed using relatively short C-terminal SAS-5 peptides (*Qiao et al., 2012*; *Hilbert et al., 2013*), as full-length SAS-5 proved difficult to purify. Our results show that SAS-5 aggregates in vitro due to a ~20 amino

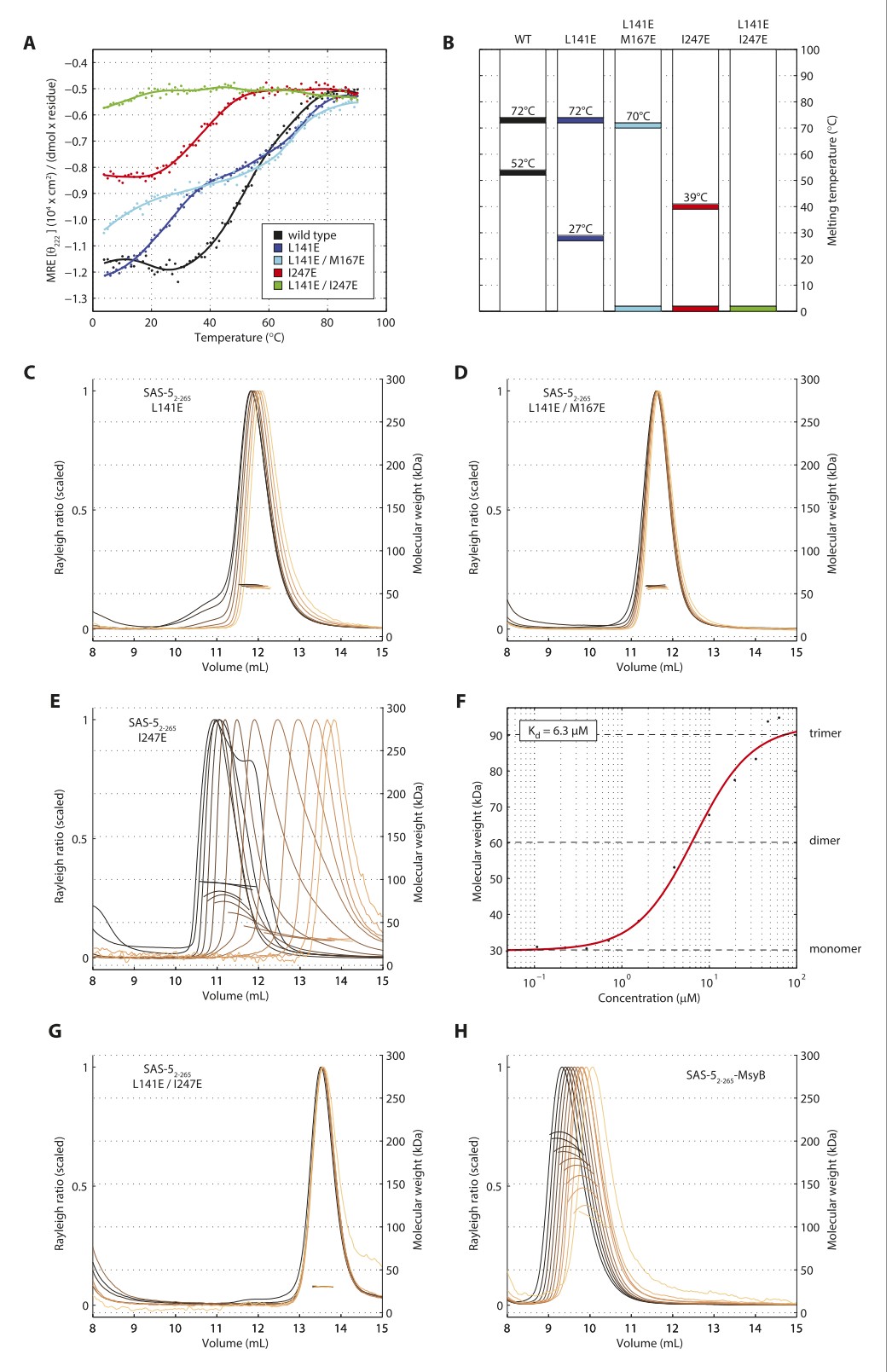

**Figure 5**. Sequential disruption of SAS-5 oligomeric interfaces. (**A**) Thermal unfolding profiles monitored by CD of SAS-5$_{2-265}$ wild-type and mutants that disrupt the coiled coil (L141E, L141E/M167E) or the *Implico* (I247E) oligomerization interface. (**B**) Graphical representation of the melting transition temperatures observed in each
*Figure 5. continued on next page*

*Figure 5. Continued*

SAS-5$_{2-265}$ variant. (**C,D**) Overlay of SEC-MALS chromatograms of SAS-5$_{2-265}$ L141E (**C**) or L141E/M167E (**D**) variants in multiple concentrations. (**E**) Similar SEC-MALS analysis of SAS-5$_{2-265}$ I247E and (**F**) plot of average molecular weight from SEC-MALS analysis of this variant as a function of on-column protein concentration. The apparent K$_d$ and molecular sizes of monomers, dimers and trimers are indicated. (**G,H**) Overlay of SEC-MALS chromatograms from the SAS-5$_{2-265}$ L141E/I247E variant (**G**), and the MsyB-tagged SAS-5$_{2-265}$ wild-type as reference (**H**).

The following figure supplement is available for figure 5:

**Figure supplement 1**. Single amino acid mutations disrupt the SAS-5$_{CC}$ and SAS-5$_{Imp}$ domains.

acid segment that may form a β-strand (*Figure 1—figure supplement 2*, *Figure 1—figure supplement 3* and *Figure 1—figure supplement 4*); this site is outside the SAS-5 region implicated in SAS-6 binding (*Qiao et al., 2012*; *Hilbert et al., 2013*; *Lettman et al., 2013*). Although we cannot exclude the possibility that SAS-5 multimerization through this predicted β-strand is physiologically

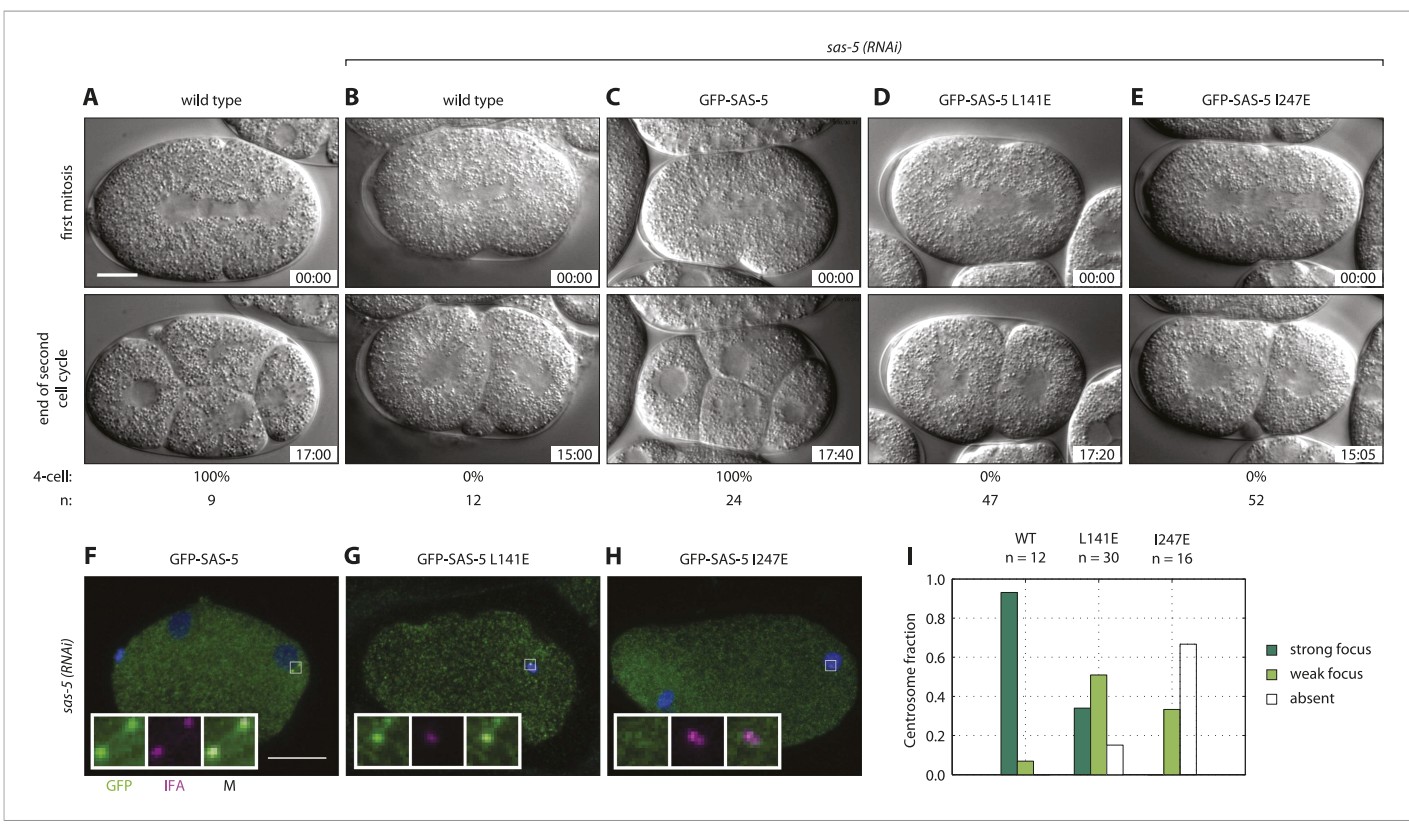

**Figure 6**. Both SAS-5 oligomerization interfaces are essential for centriole duplication in *C. elegans* embryos. Wild type (**A,B**), *gfp::sas-5* (**C**), *gfp::sas-5[L141E]* (**D**), and *gfp::sas-5[I247E]* (**E**) adult worms were subjected to *sas-5(RNAi)* or left untreated (**A**). The resulting embryos were imaged by time-lapse DIC microscopy (**A–E**, see corresponding *Videos 1–5*) or fixed and analyzed by immunofluorescence (**F–H**). (**A–E**) Embryos at the onset of cleavage furrow ingression (top) and at the end of the second cell cycle (bottom); time stamp in mm:ss from the beginning of cleavage furrow ingression at the first mitosis. The percentage of embryos that reach the four-cell stage is shown below each image, along with the number of embryos (n) analyzed per condition. (**F–H**) Embryos stained with antibodies against GFP (green) and the centriolar marker IFA (magenta); DNA is visualized in blue. Scale bar 10 μm. Insets are fivefold enlargements of boxed regions. (**I**) Centrosomal GFP localization was scored in one-cell stage embryos as shown in panels **F–H** as absent, weak focus or strong focus.

The following figure supplement is available for figure 6:

**Figure supplement 1**. GFP-SAS-5 and mutants are expressed, and resistant to sas-5 RNAi.

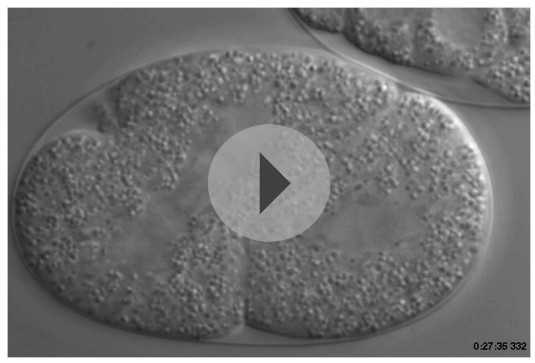

**Video 1.** Time-lapse DIC microscopy of *C. elegans* embryo over the first two cell cycles. Untreated wild type embryo. Frames were captured every 5 s and the movie is played at 12 frames/s. Embryo is oriented with anterior to the left and posterior to the right; elapsed time is shown in minutes and seconds.

relevant, we do not favor such interpretation. The aggregates formed by full-length SAS-5, as visualized by EM, are proportional in size to *C. elegans* centrioles even at very low protein concentrations (*Figure 1—figure supplement 2D*). Thus, if present in vivo, the large size of such aggregates would leave little, if any, space for other centriolar components, including SAS-6, SAS-4 and microtubules. Rather, we postulate that in vivo SAS-5 forms a complex with a yet unidentified partner using the ~20 amino acid segment identified here, and that this complex protects SAS-5 from aggregation in the cell. It is intriguing to speculate whether Protein Phosphatase 2A, which binds SAS-5 in vivo and controls SAS-5 levels (*Song et al., 2011*; *Kitagawa et al., 2011a*), may interact with SAS-5 in this region.

Structural and biophysical analysis suggests that SAS-5 can form tight dimers through its *Implico* domain, a novel dimerization interface with no known homologs outside SAS-5 proteins (*Figure 3*). These dimers cannot be disrupted even when diluting the protein to concentrations as low as 20 nM (*Figure 4A–D*). By comparison, the dimer mediated by the SAS-6 coiled coil is approximately 50-times weaker (*Kitagawa et al., 2011b*), yet the human orthologue HsSAS-6 readily forms dimers in cells (*Keller et al., 2014*). By extension, it appears reasonable to assume that SAS-5 forms dimers in the cytoplasm of *C. elegans* embryos. Such dimers would associate with SAS-6 (*Delattre et al., 2004*; *Qiao et al., 2012*; *Hilbert et al., 2013*; *Lettman et al., 2013*) and possibly SAS-4 (*Cottee et al., 2013*; *Hatzopoulos et al., 2013*), given the moderate interaction strengths of these associations. Our work shows that formation of these tight SAS-5 dimers through the *Implico* domain is necessary for SAS-5 localization to centrioles (*Figure 6H,I*). This suggests that the localization mechanism either recognizes the *Implico* domain itself, or depends on a protein association that needs two copies of a distinct SAS-5 epitope, and hence the SAS-5 dimer. Multivalent protein interactions increase the apparent strength (avidity) of association, and are common in biology (*Mammen et al., 1998*). It is possible that the interaction of SAS-5 with SAS-6, known to be necessary for localization of both proteins to centrioles (*Delattre et al., 2004*; *Qiao et al., 2012*; *Lettman et al., 2013*), benefits from such an avidity effect. It will be interesting to determine whether the L141E and I247E SAS-5 mutants prevent SAS-6 recruitment to the nascent

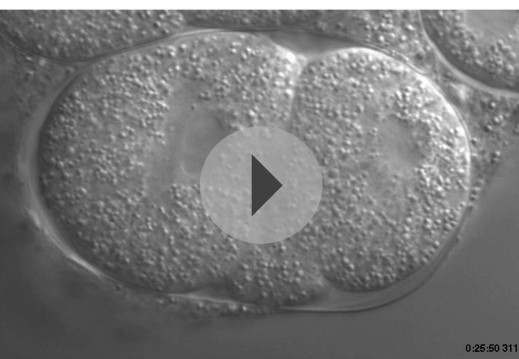

**Video 2.** Time-lapse DIC microscopy of *C. elegans* embryo over the first two cell cycles. Wild type embryo subjected to *sas-5* (*RNAi*). Frames were captured every 5 s and the movie is played at 12 frames/s. Embryo is oriented with anterior to the left and posterior to the right; elapsed time is shown in minutes and seconds.

centriole assembly site or rather prevent SAS-6 from being organized properly thereafter. In either case, this would likely hinder downstream recruitment of SAS-4, and therefore centriole formation (*Dammermann et al., 2004*; *Delattre et al., 2004*; *Qiao et al., 2012*; *Lettman et al., 2013*).

SAS-5 concentrates at centrioles, as illustrated by well-defined GFP foci of this protein (*Figure 6F,I*), and we propose that SAS-5 dimers associate further to form higher-order assemblies there. Our SEC-MALS results show that SAS-5 forms tetramers already at 1 µM concentration and a mixture of tetramers and hexamers at 10 µM (*Figure 4B*). Although the SAS-5 architecture, with two independent oligomerization interfaces, could theoretically sustain the formation of open-ended protein chains or networks, we believe it is more likely that higher-order SAS-5 assemblies cap at a hexameric size. Such hexamers, formed

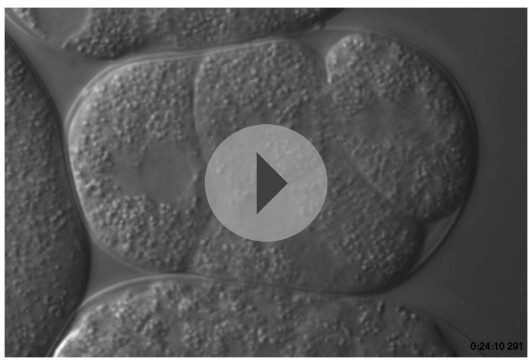

**Video 3.** Time-lapse DIC microscopy of *C. elegans* embryo over the first two cell cycles. Embryo expressing GFP-SAS-5 treated with *sas-5* (*RNAi*). Frames were captured every 5 s and the movie is played at 12 frames/s. Embryo is oriented with anterior to the left and posterior to the right; elapsed time is shown in minutes and seconds.

through the sequential association of three SAS-5 dimers (*Figure 7*), should be both kinetically and entropically favored compared to open-ended SAS-5 chains. Hexamer breakdown would require simultaneous disruption of two SAS-5 coiled coils, a rarer event compared to the disruption of a single coiled coil in an open-ended SAS-5 oligomer. Further, in the sequential assembly model of *Figure 7*, once the first coiled-coil sequence of a SAS-5 dimer has associated into a folded coiled-coil structure, the degrees of freedom of the second SAS-5 coiled-coil sequence would be reduced, and thus its association into the second folded coiled-coil structure would become entropically favored.

Disruption of these SAS-5 coiled-coil assemblies by amino acid substitutions (L141E) weakens localization to centrioles (*Figure 6G,I*), possibly by impacting the SAS-6 avidity mechanism postulated above. Intriguingly, our data suggest that formation of higher-order SAS-5 assemblies is necessary for function beyond localization, since the GFP-SAS-5 L141E mutant does not rescue centriole duplication upon depletion of endogenous SAS-5, even though it localizes to centrioles in ~1/3 of embryos (*Figure 6D,G,I*). Although some GFP-SAS-5 L141E may be provided from centrioles contributed by the sperm where endogenous SAS-5 is present, we anticipate that any paternal SAS-5 will rapidly exchange with the much larger maternal protein pool (*Delattre et al., 2004*). Thus, we expect that GFP-SAS-5 L141E visualized at centrioles (*Figure 6G,I*) is the maternal protein localized there.

The suggestion that higher-order SAS-5 assemblies may be needed for a function other than localization would imply that this protein is not a simple transporter of SAS-6, but that it acts in an additional, more structural, capacity, as has been suggested elsewhere (*Qiao et al., 2012*; *Lettman et al., 2013*). However, FRAP experiments show rapid shuttling of SAS-5 between centrioles and the cytoplasm (*Delattre et al., 2004*), which seemingly conflicts with such a postulated structural role. How can these observations be reconciled?

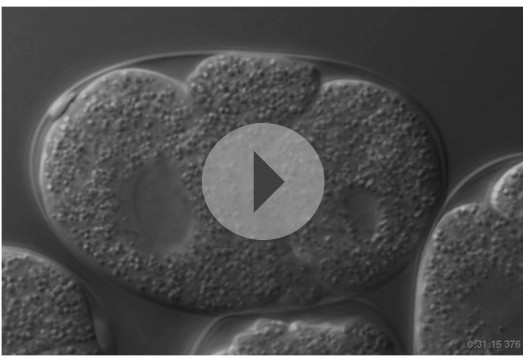

**Video 4.** Time-lapse DIC microscopy of *C. elegans* embryo over the first two cell cycles. Embryo expressing GFP-SAS-5 L141E treated with *sas-5* (*RNAi*). Frames were captured every 5 s and the movie is played at 12 frames/s. Embryo is oriented with anterior to the left and posterior to the right; elapsed time is shown in minutes and seconds.

We speculate that there may exist two different and non-exclusive mechanisms by which large SAS-5 assemblies could assist SAS-6 organization; firstly by providing multivalent SAS-6 attachment sites, and secondly by providing a structural framework. Such SAS-5 assemblies may be transiently located at the site of centriole assembly or, alternatively, could correspond to only a small fraction of total SAS-5, which would have thus gone undetected in FRAP experiments (*Delattre et al., 2004*). SAS-5 assemblies could bind multiple SAS-6 dimers and bring them together in a confined space (*Figure 8*). ZYG-1, which interacts with SAS-6 independently of SAS-5 and is also located at the duplication site (*Lettman et al., 2013*), may provide additional enhancement of local SAS-6 concentration. As the interaction affinity for the assembly of SAS-6 oligomers is weak ($K_d \sim$ 100 µM, [*Kitagawa et al., 2011b*]) this increase in local SAS-6 concentration may provide a trigger for SAS-6 oligomerisation. Furthermore, if SAS-5 forms

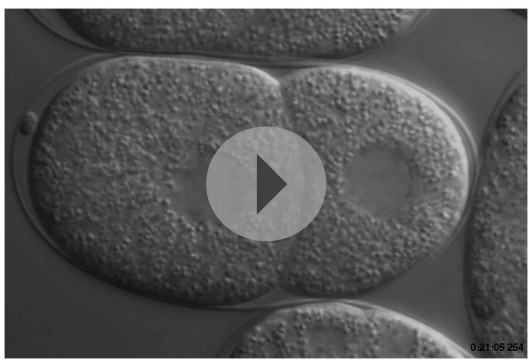

**Video 5.** Time-lapse DIC microscopy of *C. elegans* embryo over the first two cell cycles. Embryo expressing GFP-SAS-5 I247E treated with *sas-5* (*RNAi*). Frames were captured every 5 s and the movie is played at 12 frames/s. Embryo is oriented with anterior to the left and posterior to the right; elapsed time is shown in minutes and seconds.

defined hexamers as argued above, these hexamers could recruit up to six SAS-6 dimers (*Qiao et al., 2012*; *Hilbert et al., 2013*) at the initiation site of SAS-6 oligomerisation. This number of SAS-6 moieties would be sufficient for just over one turn of the proposed SAS-6 spiral (*Hilbert et al., 2013*), but not for a SAS-6 ring (*Kitagawa et al., 2011b*; *Van Breugel et al., 2011*; *Van Breugel et al., 2014*). Thus, a hexameric SAS-5 architecture would provide an initial structural guide for the configuration of the SAS-6 oligomer. This would be lost in SAS-5 mutants that disrupt its architecture or its interaction with SAS-6 (*Delattre et al., 2004*; *Qiao et al., 2012*; *Lettman et al., 2013*), therefore explaining the absence of centriole formation.

Our model is conceptually similar to that proposed, albeit in the absence of structural data, for the vertebrate analogue of SAS-5, STIL (*Ohta et al., 2014*), and underlies the possible functional similarities between these two proteins. Although the oligomeric status of STIL has not been resolved yet, that of the *Drosophila* analogue of SAS-5 and STIL, Ana2 (*Stevens et al., 2010*), suggests the presence of higher-order assemblies possibly formed through interactions with the dynein light chain (*Wang et al., 2011*; *Slevin et al., 2014*). However, the specific structural features of SAS-5 are likely restricted to close homologs of this protein, as we were unable to identify sequences homologous to the SAS-5 coiled-coil and *Implico* domains in either STIL or Ana2. Further, no SAS-5 homolog has been identified in *Trichinella spiralis*, a nematode that may feature SAS-6

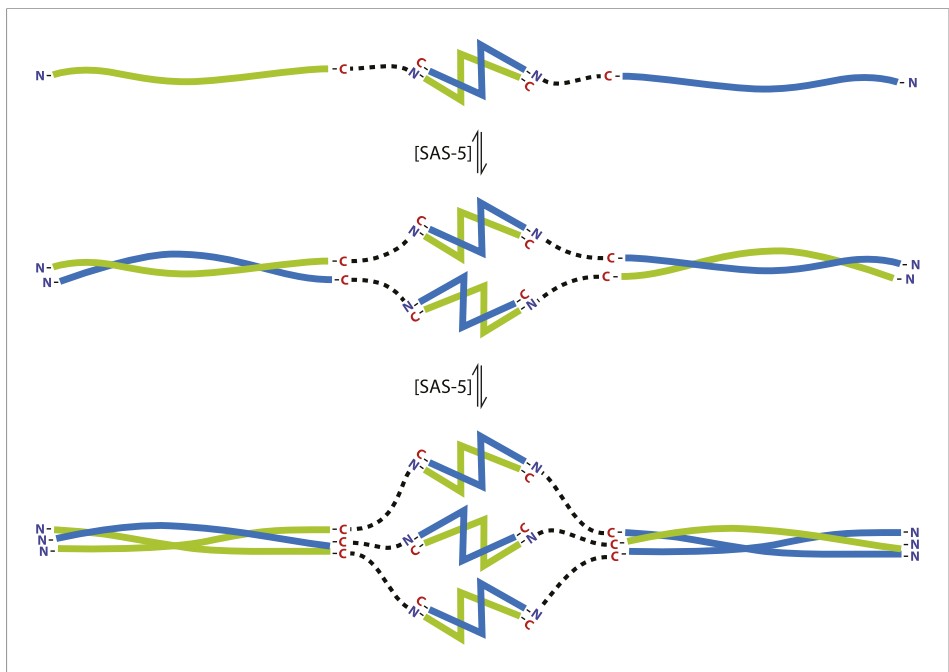

**Figure 7.** The SAS-5 architecture allows formation of higher oligomers. Schematic representation of SAS-5 oligomerization, showing the coiled-coil and *Implico* domains, without the disordered N- and C-termini of this protein. Tight dimers of SAS-5, mediated by the *Implico* domain, could come together through the coiled coil to form, first, a tetramer and ultimately a hexamer in a concentration-dependent manner. This oligomerisation model does not take into account protein aggregation mediated by the predicted β-strand at the SAS-5 C-terminus.

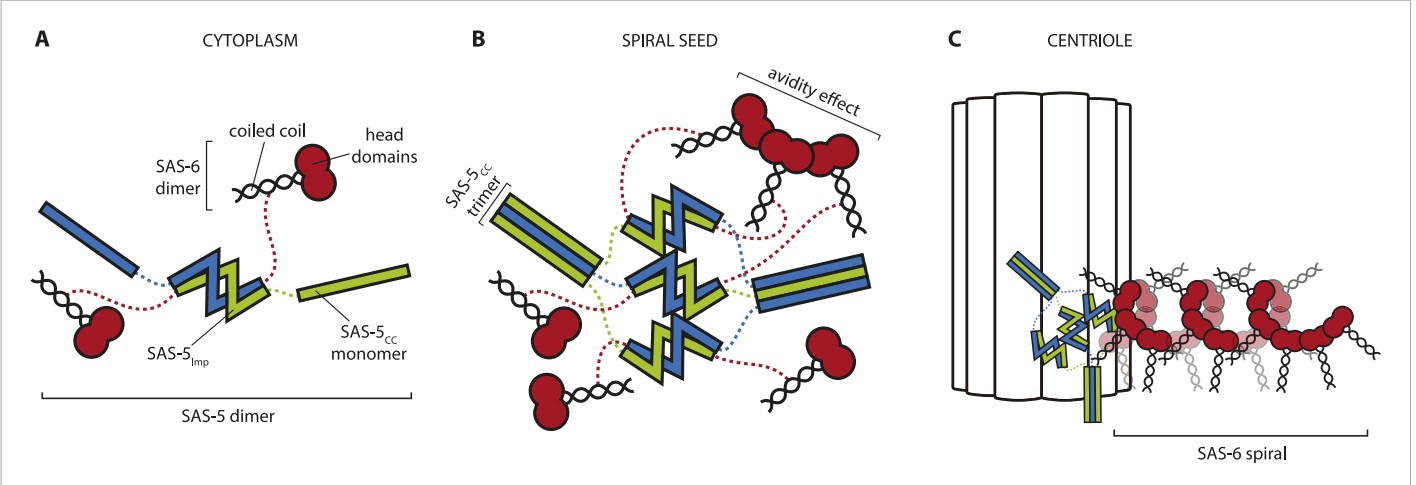

**Figure 8**. Hypothetical mechanism of SAS-5 function. (**A**) SAS-5 forms tight dimers in the cytoplasm, and it may also associate with SAS-6 dimers depending on the relative levels of each protein. (**B**) SAS-5 localizes at the centriole, where it forms higher-order assemblies. These assemblies bring together multiple SAS-6 dimers in close proximity to one another, thereby increasing the local SAS-6 concentration. Thus, SAS-5 assemblies create a seeding point from which the SAS-6 oligomer forms (**C**); the first step in creating the *C. elegans* central tube.

cartwheels, rather than spirals, at its centrioles (*Hilbert et al., 2013*). Thus, SAS-5 may have evolved in tandem with SAS-6 in order to accommodate the structural divergence of the latter protein and promote spiral formation. Comparison of the SAS-5 structure with those of Ana2 and STIL will be informative in assessing the extent to which the molecular architecture of these proteins is conserved.

In summary, we report here a combined structural and functional analysis of SAS-5 architecture, which underlies the role of large SAS-5 assemblies in centriole duplication in *C. elegans* embryos. We propose that SAS-5 promotes assembly of SAS-6 into oligomers by providing a multivalent structural framework.

## Note added in proof

Cottee, Muchalik et al. (*Cottee et al., 2015*) have now shown that Drosophila Ana2 and vertebrate STIL, the functional equivalents of SAS-5, can also form higher-order oligomers that are essential for function. Both Ana2 and STIL tetramerize through their coiled-coil domains. It remains to be determined whether the different SAS-5 and Ana2/STIL architectures reflect a divergence in centriole architecture between these organisms.

## Materials and methods

### Construction of protein expression plasmids

We have constructed a line of entero-bacterial expression vectors, called pFloat, derived from pET30a (Novagen, Madison, WI) and engineered to include N-terminal His$_6$- and solubility tags, and a human rhinovirus 3C protease cleavage sequence. To accelerate construct screening we further inserted a ccdB promoter and gene cassette, amplified from pDONR211 (Invitrogen, Grand Island, NY), inside the multi-cloning site. Various solubility tags included thioredoxin (TRX), glutathione S-transferase (GST), S-tag, colicin E9 immunity protein (Im9), maltose-binding protein (MBP), small ubiquiting-like modifier (SUMO), haloalkane dehalogenase (HALO), trigger factor (TF), N utilization substance protein A (NusA), acidic protein MsyB, yellow fluorescent protein (YFP) and enhanced green fluorescent protein (eGFP); the solubility tag sequences were amplified from the pOPIN (*Bird, 2011*) and pGEX (GE Healthcare, Little Chalfont, UK) vector systems, or directly from the *Escherichia coli* strain BL21 genome. All amplifications were performed using high fidelity Phusion polymerase (New England Biolabs, Ipswich, MA), and vectors were constructed using seamless cloning methods (Gibson Assembly, New England Biolabs).

### Protein expression and purification

SAS-5 gene fragments from *C. brenneri*, *C. remanei*, *C. sinica* and *C. tropicalis* were synthesised by IDT, Coralville, IA. *C. briggsae* SAS-5 was amplified from genomic DNA (gifted by Marie-Anne

Félix, Ecole Normale Supérieure). Full-length or gene fragments encoding for SAS-5 constructs were cloned in pFloat vectors, transformed to *E. coli* strain BL21(DE3) Rosetta2 (Novagen) and grown at 37°C in lysogeny broth (LB) media. Protein expression was induced at $OD_{600}$ 0.6 by isopropyl-β-D-1-thiogalactopyranoside (IPTG, 0.5 mM final concentration) and lowering the temperature to 18°C. After ∼16 hr incubation cells were harvested and resuspended in PBS solution (10 mM $Na_2HPO_4$, 1.8 mM $KH_2PO_4$, 137 mM NaCl, 2.7 mM KCl, pH 7.4) supplemented with 350 mM NaCl, 0.1 mM phenylmethanesulfonyl fluoride (PMSF), a protease inhibitor cocktail (cOmplete, Roche, Basel, Switzerland), 0.25 mg/ml lysozyme and 10 μg/ml DNAse I. Resuspended cells were disrupted by sonication while chilled on ice, and cell lysate was cleared by centrifugation at 40,000×*g* and 0.22 μM filtration. Imidazole at 20 mM final concentration was added to clarified lysates prior to incubation with TALON (GE Healthcare) metal affinity resin pre-equilibrated with PBS supplemented with 350 mM NaCl and 20 mM imidazole. Resin beads were washed with PBS supplemented with 350 mM NaCl, 1 mM β-mercaptoethanol and 20 mM imidazole, and protein was eluted in PBS supplemented with 350 mM NaCl, 1 mM β-mercaptoethanol and 500 mM imidazole.

Further purification of proteins was performed at 4°C by anion- (HiTrap Q HP, GE Healthcare) or cation-exchange (HiTrap SP HP, GE Healthcare) chromatography in 10 mM Tris–HCl or $Na_2HPO_4$, 50 mM NaCl, 3 mM β-mercaptoethanol, pH 8.0 (for anion-exchange) or pH 6.5 (for cation-exchange) running buffer, and proteins were eluted using a NaCl gradient to 2 M. $His_6$- and solubility-tags were cleaved, whenever necessary, by overnight incubation with Prescission protease (GE Healthcare); proteins were further purified by size-exclusion chromatography in HiLoad Superdex 75, Superdex 200 or Superose 6 columns (GE Healthcare) equilibrated in 10 mM Tris–HCl, 300 mM NaCl, 2 mM β-mercaptoethanol, pH 7.5 buffer. Pure protein fractions were buffer-exchanged by dialysis, and concentrated by centrifugal ultrafiltration. Protein concentration was estimated by UV absorption at 280 nm, and protein identity confirmed by electrospray ionization mass-spectrometry.

## Biophysical analysis

Far-UV CD spectra (260–180 nm) were measured using a Jasco (Easton, MD) J-815 Spectropolarimeter at 10°C. Protein samples were exchanged to a 10 mM $NaH_2PO_4$, 100 mM NaF, 1 mM β-mercaptoethanol, pH 7.5 buffer. CD sample concentrations are shown in *Table 3*. Data were truncated at high tension voltage exceeding 800 V. After buffer background subtraction, data were smoothed using a Savitzky–Golay filter (*Savitzky and Golay, 1964*). CD spectra were deconvoluted using the DICHROWEB server (*Whitmore and Wallace, 2008*). Measurements of thermal stability monitored CD signal at 222 nm while temperature was increased by 1°C/min between 4°C and 90°C.

SEC-MALS analysis was performed at 20°C using analytical Superdex 75, Superdex 200 or Superose 6 columns (GE Healthcare) and a Shimadzu (Kyoto, Japan) chromatography system, connected in-line to a Heleos8+ multi-angle light scattering detector and an Optilab T-rEX refractive index (RI) detector (Wyatt Technologies, Goleta, CA). A 1:1 dilution series of protein samples in 10 mM Tris–HCl, 300 mM NaCl, 2 mM β-mercaptoethanol, pH 7.5 buffer were injected in this system, and the resulting MALS, RI and UV traces processed in ASTRA 6 (Wyatt Technologies). On-column protein concentration was calculated from the differential RI, assuming dn/dc of 0.1850 ml/g. SEC-MALS data were fit with a two-state model to derive association parameters, and the resulting $K_d$ estimated from the inflection point of the fit. Details of SEC-MALS sample concentrations, monomeric molecular weight and column used are shown in *Table 3*.

Native mass-spectrometry was performed on protein samples in 250 mM $NH_4Acetate$, pH 7.5 buffer using a modified Synapt G1 High Definition Mass Spectrometry Quadrupole Time of Flight instrument (Waters, Milford, MA) (*Bush et al., 2010*) calibrated using 10 mg/ml CsI. 2 μl aliquots of sample were delivered by nano-electrospray ionization via gold-coated capillaries, prepared in house (*Hernandez and Robinson, 2007*). Instrumental parameters were as follows: source pressure 6.0 mbar, capillary voltage 1.20 kV, cone voltage 50 V, trap energy 10 V, bias voltage 5 V and trap pressure 0.0163 mbar.

For negative stain electron microscopy $SAS-5_{FL}$ protein samples (5 μl, 0.1 mg/ml) in 10 mM Tris–HCl, 300 mM NaCl, 2 mM β-mercaptoethanol, pH 7.5 buffer were applied to glow-discharged homemade carbon-coated copper mesh grids and allowed to adsorb for 1 min. After blotting off any excess liquid, negative stain was applied by adding 3 μl of 1% wt/vol uranyl acetate to the grid. Blotting and staining was repeated 2 times, and the grids were then allowed to air dry for 10 min. The grids were imaged in a JEOL (Tokyo, Japan) JEM-2010 electron microscope equipped with a high brightness $LaB_6$ filament

**Table 3**. CD and SEC-MALS experimental details

**CD**

| Protein name | Concentration (µM) |
|---|---|
| SAS-5$_{FL}$ | 1.5 |
| SAS-5$_{\Delta282-295}$ | 3 |
| SAS-5$_{FLEX}$ | 3 |
| SAS-5$_{2-265}$ | 6 |
| SAS-5$_{2-265}$ L141E | 6 |
| SAS-5$_{2-265}$ L141E/M167E | 6 |
| SAS-5$_{-265}$ I247E | 6 |
| SAS-5$_{2-265}$ L141E/I247E | 6 |
| SAS-5$_{125-265}$ | 10 |
| SAS-5$_{CC}$ | 20 |
| SAS-5$_{CC}$ L141E | 20 |
| SAS-5$_{Imp}$ | 20 |
| SAS-5$_{Imp}$ I247E | 20 |
| SAS-5 N-terminus | 20 |
| SAS-5 C-terminus | 5 |

**SEC-MALS**

| Protein name | Monomeric MW (kDa) | Size exclusion column | Minimum concentration (µM)* | Maximum concentration (µM)* |
|---|---|---|---|---|
| SAS-5$_{FL}$ | 46 | Superose6 | 0.10 | 0.57 |
| SAS-5$_{FLEX}$-MsyB | 62 | Superose6 | 0.03 | 9.5 |
| SAS-5$_{2-265}$-MsyB (*Figure 4*) | 46 | Superose6 | 0.02 | 41 |
| SAS-5$_{2-265}$-MsyB (*Figure 5*) | 46 | Superdex200 | 0.05 | 41 |
| SAS-5$_{2-265}$ L141E | 46 | Superdex200 | 1.32 | 63 |
| SAS-5$_{2-265}$ L141E/M167E | 46 | Superdex200 | 1.16 | 63 |
| SAS-5$_{2-265}$ I247E | 46 | Superdex200 | 0.05 | 63 |
| SAS-5$_{2-265}$ L141E/I247E | 46 | Superdex200 | 0.82 | 63 |
| SAS-5$_{Imp}$ | 6.5 | Superdex75 | 2.9 | 292 |
| SAS-5$_{CC-L}$ | 12.5 | Superdex75 | 0.18 | 152 |
| *C. briggsae* SAS-5 (96–199) | 12.7 | Superdex75 | 0.10 | 149 |
| *C. brenneri* SAS-5 (1–82) | 10.2 | Superdex75 | 0.21 | 186 |
| *C. remanei* SAS-5 (98–206) | 13 | Superdex75 | 0.21 | 146 |
| *C. sinica* SAS-5 (94–204) | 13 | Superdex75 | 0.16 | 145 |
| *C. tropicalis* SAS-5 (94–199) | 12.5 | Superdex75 | 0.12 | 152 |

*Refers to on-column concentration calculated from the protein differential refractive index.

and an ORIUS SC1000 (Model 832) camera (Gatan, Abingdon, UK). Images were acquired using an accelerating voltage of 200 kV, at a nominal magnification of 16,000× to 52,000×.

## Crystallization, crystallographic data collection and refinement

Crystals were obtained using the sitting drop vapor diffusion technique. A Mosquito robot (TTP LabTech, Melbourn, UK) was used to setup 200 nL-size drops with 1:1 and 1.3:0.7 ratios of

protein to mother liquor. For SAS-5$_{CC}$, protein in 10 mM Tris–HCl pH 7.5, 200 mM NaCl buffer and at concentration of 20 mg/ml was mixed with 0.1 M MOPS/HEPES-Na pH 7.5, 12.5% wt/vol PEG 1000, 12.5% wt/vol PEG 3350, 12.5% wt/vol MPD, 0.03 M NaNO$_3$, 0.03 M Na$_2$HPO$_4$, 0.03 M (NH$_4$)$_2$SO$_4$. Crystals developed at 4°C in 30 days, were flash-cooled in liquid nitrogen and diffracted to a maximum resolution of 1.73 Å at the Diamond Light Source (DLS, Harwell, UK) beamline I04. The space group was determined as P2$_1$2$_1$2$_1$ with eight molecules per asymmetric unit. Crystallographic data (statistics in *Table 1*) were integrated in MOSFLM (*Leslie and Powell, 2007*) or XDS (*Kabsch, 2010*) and scaled by Aimless (*Evans and Murshudov, 2013*) or XSCALE (*Kabsch, 2010*). Scaling reveals substantial diffraction anisotropy, which was corrected using the Diffraction Anisotropy Server (UCLA Molecular Biology Institute, *Strong et al., 2006*) whilst applying a high resolution cutoff of 1.8 Å. Statistics of the anisotropically corrected data are shown in *Table 2*. Phase information for SAS-5$_{CC}$ was obtained from a highly redundant dataset with maximum resolution of 1.9 Å collected on Pb-derivatized crystals at a wavelength of 0.947 Å at the DLS beamline I03. For Pb-derivatization the native SAS-5$_{CC}$ crystals were incubated with 1 mM trimethyl lead (IV) acetate for 16 hr prior to cooling. Phasing by SAD was performed using PHENIX.autosol (*Adams et al., 2002*) which located and refined 9 Pb sites to produce a density map with initial figure of merit of 0.36. Initial model building was done with PHENIX.autobuild (291 residues built, 222 identified out of 427 residues in the asymmetric unit). Iterative model building with COOT (*Emsley and Cowtan, 2004*) and refinement against the anisotropically corrected native data in Buster 2.10 (*Bricogne et al., 2011*) using automatic NCS and TLS restraints (*Smart et al., 2012*) yielded the final SAS-5$_{CC}$ model.

For SAS-5$_{Imp}$, protein in 10 mM HEPES pH 7.5, 150 mM NaCl, 1 mM DTT buffer and at concentration of 10 mg/ml was mixed with 0.1 M MES/imidazole pH 6.4, 10% wt/vol PEG 20,000, 20% vol/vol PEG MME 550, 0.1 M carboxylic acids mixture (*Gorrec, 2009*). Crystals developed at 18°C in 3 days, were cryo-protected by brief immersion to mother liquor supplemented with 20% vol/vol glycerol, flash-cooled in liquid nitrogen and diffracted to 1.0 Å at the European Synchrotron Radiation Facility (ESRF, Grenoble, France) beamline ID14-4. The space group was determined as P12$_1$1 with two molecules per asymmetric unit. Data integration and scaling was performed as in SAS-5$_{CC}$ (*Table 1*), however no anisotropy correction was required. Phase information was obtained from a highly redundant dataset with maximum resolution of 1.7 Å collected on Hg-derivatized crystals at a wavelength of 1.007 Å at the DLS beamline I02. For Hg-derivatization native SAS-5$_{Imp}$ crystals were incubated with 2 mM mercury (II) acetate for 16 hr prior to cooling. Phasing by SAD was performed using PHENIX.autosol (*Adams et al., 2002*) which located and refined 2 Hg sites to produce a density map with initial figure of merit of 0.39. Initial model building was done with PHENIX.autosol (86 residues built and 51 identified out of 116 residues in the asymmetric unit). Iterative model building with COOT (*Emsley and Cowtan, 2004*) and refinement against the native data in PHENIX.refine (*Adams et al., 2002*) using explicit hydrogens and atom-level anisotropic restraints.

Model quality was assessed by MolProbity (*Chen et al., 2010*). For graphical representation, we used UCSF Chimera (*Pettersen et al., 2004*). Initial crystal inspection and data collection took place at the DLS beamline I04-1.

## MD simulations

All MD simulations were performed using GROMACS 4 (*Pronk et al., 2013*) and the GROMOS96 53a6 force field (*Oostenbrink et al., 2004*). The crystal structure of SAS5$_{CC}$ was placed in a cubic simulation box of 1452 nm$^3$ volume with periodic boundary conditions. The protein was solvated by adding 10,637 explicit SPC water molecules. Na$^+$ and Cl$^-$ ions were added to the final concentration of 0.1 M. The system was first minimized with the steepest descend algorithm, equilibrated in constant temperature (NVT, 300 K) for 100 ps, and followed by equilibration in constant temperature and pressure (NPT, 1 bar) for 500 ps. The pressure was controlled with an isotropic Parrinello-Rahman barostat (*Parrinello and Rahman, 1981*) applied to the entire system with a time constant of 2.0 ps and compressibility of $4.5 \times 10^{-5}$ bar$^{-1}$. The temperature was controlled with two velocity rescaling thermostats (*Berendsen et al., 1984*) applied to the protein and solvent, with a time constant of 0.1 ps. Electrostatic interactions were calculated using the particle-mesh Ewald (*Darden et al., 1993*) summation method. Ten replicate simulations of 45 ns duration were performed in the Oxford Advanced Research Computing facility.

## Nematode strains and RNA interference

The genomic sequence of *sas-5* (Wormbase, F35B12.5a) was recoded for resistance to RNAi directed against the endogenous gene, and synthesized by Genscript (Piscataway, NJ). Intronic sequences were kept intact, as well as the bases neighboring the splice sites: 3 bases at 3′ ends and 6 bases at 5′ ends of exons. The codon adaptation index was preserved at the level of the wild type (WT: 0.777, recoded: 0.712). The recoded gene was amplified and cloned in pIC26 using the SpeI restriction site. pIC26 contains a *pie-1* promoter and 3′UTR, as well as a GFP coding sequence fused upstream of and in frame with *sas-5* (*Cheeseman and Desai, 2005*). *C. elegans* culture was performed according to standard procedures (*Brenner, 1974*). GFP–SAS-5 transgenic animals were generated by bombardment (*Praitis et al., 2001*). Integrated lines were recovered after bombardment with *gfp::sas-5 L141E* (strain GZ1301; isIs51{*pie-1::gfp:sas-5[recL141E]*}) and *I247E* (strain GZ1302; isIs52{*pie-1::gfp:sas-5[recI247E]*}), but GFP-SAS-5 bombardment led to the recovery of non-integrated lines. For these experiments a line with <50% transmission of the extra chromosomal array was used (strain GZ1300; isEx5{*pie-1::gfp:sas-5 [rec]*}). RNAi was carried out by selecting L3–L4 wild-type hermaphrodites and feeding them for 26 hr at 24°C using the *sas-5(RNAi)* feeding strain, which targets nucleotides 301–1170 of the genomic sequence (*Delattre et al., 2004*).

## Antibodies and immunofluorescence

Embryos were prepared for immunofluorescence essentially as described (*Gönczy et al., 1999*), with a fixation time of 2 min in −20°C methanol and an additional blocking step in PBS, 2% wt/vol BSA before antibody incubation. Primary and secondary antibodies were diluted in PBS as follows: rabbit anti-GFP (gift from Viesturs Simanis, EPFL, 1/500), mouse anti-IFA (*Leung et al., 1999*, 1/50), goat anti-GFP (Abcam, Cambridge, UK, ab6673, 1/250), goat anti-rabbit Alexa Fluor 488, goat anti-mouse Alexa Fluor 568, donkey anti-goat Alexa Fluor 488, donkey anti-mouse Alexa Fluor 568 (Life Technologies, Grand Island, NY, all 1/500). Primary antibodies were incubated overnight at 4°C and secondary incubations were at room temperature for 1 hr. 1 µg/ml Hoechst 33,258 (Sigma-Aldrich, St. Louis, MO) was used to visualize DNA.

## Microscopy

Time-lapse DIC microscopy of early embryos was carried out as described (*Gönczy et al., 1999*), recording one image every 5 s at 24°C. A Zeiss (Oberkochen, Germany) LSM 700 confocal microscope was used for fluorescence microscopy. Z-sections were imaged at an interval of ~0.3 µm. Fluorescent images in *Figure 6F–H* are maximum intensity projections. Dual time-lapse and fluorescence DIC imaging was performed on a Zeiss Axioplan 2 (*Bellanger and Gönczy, 2003*). The motorised filter wheel, two external shutters, and the 1392 × 1040 pixel, 12-bit Photometrics CoolSNAP ES2 (Photometrics, Tucson, AZ) were controlled by µManager (*Edelstein et al., 2010*). Images were acquired with an exposure time of 100 ms for the DIC and 200 ms for the GFP channel using the Zeiss Filter Set 10 (GFP). The GFP images presented in *Figure 6—figure supplement 1* are maximum intensity projections of 8 z-sections acquired with an interval of 0.5 µm. For quantification of fluorescence intensities, the integrated intensity was measured from three areas of fixed size within each embryo and then averaged. The average background fluorescence of wild-type (GFP-negative) embryos was subtracted, then values were normalized to the average value of the GFP-SAS-5 wild type intensity. All images were processed and fluorescence intensities measured using ImageJ (*Schneider et al., 2012*).

## Western blotting

Embryos were synchronized by bleaching adult worms, hatched and matured at 24°C; 50 adult worms were then collected manually, boiled in Laemmli buffer for 5 min, sonicated, and loaded on a 4–12% wt/vol SDS-PAGE (Biorad, Hercules, CA). Proteins were transferred to a 0.45 µm nitrocellulose membrane, blocked with 5% wt/vol milk powder in PBS, 0.1% vol/vol Tween-20, and blotted with rabbit anti-SAS-5 (full length, 1/1000, (*Delattre et al., 2004*)) or rabbit anti-GFP (1/1000, gift from Viesturs Simanis, EPFL), then re-blotted with mouse anti-α-tubulin (1/10,000, DM1α, Sigma-Aldrich) antibodies. Immunocomplexes were visualized using anti-rabbit or anti-mouse HRP conjugated secondary antibodies at 1/5000 dilution (Promega, Madison, WI), chemiluminescence kit (Roche), and X-ray films (Fujifilm, Tokyo, Japan).

## Sequence notes

SAS-5 protein sequences in this study correspond to UniProt database entries Q20010 (*C. elegans*), Q60M48 (*C. briggsae*), G0NH40 (*C. brenneri*) and E3LPN2 (*C. remanei*). *C. sinica* and *C. tropicalis* SAS-5 sequences were identified from the respective genomes by blast searches starting from the *C. elegans* variant. For *C. sinica* the University of Edinburgh sp. 5 genome project (Csp5_scaffold_00353) was used, while for *C. tropicalis* the Genome Institute at Washington University *Caenorhabditis* sp. 11 genome project (Csp11.Scaffold629) served as reference. The flexible linker used in SAS-5$_{FLEX}$ to replace the 'sticky' amino acid segment has the sequence SGAAGSSGAAGSSG.

## Acknowledgements

We are grateful to David Staunton and Edward Lowe for the upkeep of the Oxford Biochemistry Biophysics and Crystallization facilities, respectively, Bernhard Knapp for assistance with supercomputer access and Coralie Busso for generating transgenic lines. We thank Manuel Hilbert for assistance with SAS-5 constructs, and Marie-Anne Félix and Viesturs Simanis for reagents. We are grateful to Michel Steinmetz, Virginie Hamel, Michele Erat and Sarah Bianchi for insightful discussions and critical reading of our manuscript. We acknowledge the Diamond Light Source and the European Synchrotron Radiation Facility for provision of X-ray radiation facilities, and the Oxford Advanced Research Computing for supercomputer resources. KBR was supported by a studentship from the Engineering and Physical Sciences Research Council Centre for Doctoral Training in Systems Biology, and by a Travel Grant from Boehringer Ingelheim Fonds.

## Additional information

### Funding

| Funder | Grant reference | Author |
|---|---|---|
| Wellcome Trust | 088497/Z/09/Z | Ioannis Vakonakis |
| Biotechnology and Biological Sciences Research Council (BBSRC) | BB/J008265/1 | Ioannis Vakonakis |
| European Research Council (ERC) | AdG 233335 | Pierre Gönczy |
| European Research Council (ERC) | AdG 340227 | Pierre Gönczy |
| European Research Council (ERC) | IMPRESS, ID 268851 | Carol V Robinson |

The funders had no role in study design, data collection and interpretation, or the decision to submit the work for publication.

### Author contributions

KBR, NJD, IV, Conception and design, Acquisition of data, Analysis and interpretation of data, Drafting or revising the article; GNH, JY, Acquisition of data, Analysis and interpretation of data; SKP, Acquisition of data; CVR, CMD, Conception and design; PG, Conception and design, Analysis and interpretation of data, Drafting or revising the article

## Additional files

### Major datasets

The following datasets were generated:

| Author(s) | Year | Dataset title | Dataset ID and/or URL | Database, license, and accessibility information |
|---|---|---|---|---|
| Rogala KB, Hatzopoulos GN, Vakonakis I | 2015 | Structure of the C. elegans SAS-5 coiled coil domain | http://www.rcsb.org/pdb/explore/explore.do?structureId=4YV4 | Publicly available at RCSB Protein Data Bank (Accession No: 4YV4). |

| Author(s) | Year | Dataset title | Dataset ID and/or URL | Database, license, and accessibility information |
|---|---|---|---|---|
| Rogala KB, Hatzopoulos GN, Vakonakis I | 2015 | Structure of the C. elegans SAS-5 Implico dimerization domain | http://www.rcsb.org/pdb/search/structidSearch.do?structureId=4YNH | Publicly available at RCSB Protein Data Bank (Accession No: 4YNH). |

Standard used to collect data: Crystallographic data and structure refinement statistics have been reported according to usual (but not standardized) format. Structures have been validated by the databank and the validation documents are attached.

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
