## [Decision Letter]

Thank you for submitting your work entitled “The *Caenorhabditis elegans* protein SAS-5 forms large oligomeric assemblies critical for centriole formation” for peer review at *eLife*. Your submission has been favorably evaluated by John Kuriyan (Senior editor) and two reviewers, one of whom is a member of our Board of Reviewing Editors.

In this manuscript, Rogala et al., investigate the structural characteristics of SAS-5, a *C. elegans* protein that is essential for centriole formation. Using a variety of biochemical techniques, the authors show that SAS-5 oligomerizes via multiple domains and that SAS-5 oligomerization essential for proper centriole duplication in the *C. elegans* embryo. Furthermore, the authors determined the crystal structures of a central coiled-coil domain and a novel dimerization domain, which the authors term Implico. This study is well done and provides novel insight on the structure and function of a key centriole protein. Furthermore, the identification of a novel dimerization domain will no doubt be of considerable interest to the wider community.

We have a number of questions that we would like you to address, by further discussion in the text.

How could the flexible SAS-5 hexamer that you propose facilitate the formation of a defined SAS-6 spiral and why would oligomerisation stop with a hexamer at all? Further oligomerization could easily be explained if one considers all the other alternative arrangements of the oligomers e.g., the two coiled coil parts (from one Implico domain dimer) joining up to form a parallel dimer with a free valency to take up a third coiled coil helix (linker between Implico domain and coiled coil is long enough for that) or coiled coil helices provided by different SAS-5 dimers coming together from different sides such that one would get chains or meshworks. How do you think that the resulting amorphous and flexible assembly structurally facilitates a defined SAS-6 assembly.

You propose that SAS-5 multimerization is critical for organizing SAS-6 complexes into spirals with 9-fold symmetry (see the last sentence of the Abstract, the last sentence of the Introduction, and the Discussion). Since this hypothesis has not been tested, I feel that this statement is inappropriate for the Abstract and Introduction. More work could be done to see if purified SAS-5 affects SAS-6 assembly in vitro, but I think that is beyond the scope of this paper. However, I think speculation is appropriate for the Discussion section. Furthermore, in the Discussion section the authors should also consider the possibility that the predicted Beta sheet spanning a.a. 282–295 in SAS-5 might be physiologically relevant and promote further multimerization beyond hexamers. The authors cannot dismiss this possibility since they never tested the functionality of SAS-5_Δ282-295_ in vivo.

The model in Figure 8 is misleading since the authors never tested if SAS-5 oligomerization affects SAS-6 organization. This figure should be removed.

No reference is made to an earlier characterization of SAS-5 structure published by [65] (Worm). The authors should mention this earlier work, why their results are different, and remove the lines in the Discussion section that claim that their study is the first structural and functional analysis of SAS-5 architecture.

The MALS experiments with the coiled coil construct have not been done at higher concentrations, the use of MD analysis to demonstrate trimerisation of the coiled coil is rather weak evidence and some of the MALS runs in Figure 2 are a bit puzzling: The broad black curve gives a flat MW throughout the peak and the tail, while the brown curves that elute at that tail position are less than half that MW. If the assembly goes concentration dependent monomer-dimer-trimer then that behaviour is a bit hard to understand as the black curve should reflect these steps throughout its very broad peak.

It is not always 100% clear whether the solubility tag has been removed or left on.

CD deconvolution into secondary structure percentages tends not to be very accurate in terms of exact numbers and I would consider just mentioning that constructs where found to be mainly helical and so on.

I'm surprised that the authors didn't test if SAS-5 multimerization affects SAS-6 recruitment in vivo. The strong centriole/cell division phenotype seen in the L141E and I247E mutants might be due to the fact that not enough SAS-6 is recruited to the nascent daughter centriole, rather than SAS-6 not being organized properly once delivered there. The authors could use immunofluorescence to test this possibility.

The model in Figure 4 should be revised so that it is clear that the proposed model is for a modified version of SAS-5 that lacks the central predicted Beta sheet (SAS-5_FLEX_).

---

## [Author Response]

*How could the flexible SAS-5 hexamer that you propose facilitate the formation of a defined SAS-6 spiral and why would oligomerisation stop with a hexamer at all? Further oligomerization could easily be explained if one considers all the other alternative arrangements of the oligomers e.g., the two coiled coil parts (from one Implico domain dimer) joining up to form a parallel dimer with a free valency to take up a third coiled coil helix (linker between Implico domain and coiled coil is long enough for that) or coiled coil helices provided by different SAS-5 dimers coming together from different sides such that one would get chains or meshworks. How do you think that the resulting amorphous and flexible assembly structurally facilitates a defined SAS-6 assembly*.

We accept that our SEC-MALS results do not conclusively show that SAS-5 oligomers are limited to hexamers as opposed to open-ended chains or networks. Unfortunately, our attempts at probing the limits of SAS-5 oligomers using AUC and SAXS have not met with success due to protein precipitation and aggregation during these experiments. Formation of defined SAS-5 hexamers is thus a conjecture, and as such we have moved their mention from the Results section to the Discussion of the revised manuscript. There, we acknowledge that SAS-5 open-ended polymers are possible, but argue that SAS-5 hexamers would be kinetically and entropically more stable, compared to open-ended polymers. For the same reasons Figure 4 of the original manuscript has been re-positioned as Figure 7 of the Discussion.

SAS-5 oligomerisation would assist SAS-6 self-assembly through two different mechanisms. Firstly and primarily, SAS-5 oligomers could bind multiple SAS-6 dimers, bringing them close together in space and thereby increasing the SAS-6 local concentration. This type of avidity effect is very likely necessary for SAS-6 oligomerisation, as the self-assembly interaction strength of this protein is weak (K_d_ ∼100 μM, see Kitagawa et al. Cell, 2011). In addition to the mechanism above, if SAS-5 oligomers do not readily exceed the tetrameric/hexameric states seen in solution, this would imply a close approach of up to four or six SAS-6 dimers per SAS-5 oligomer. This number of SAS-6 moieties would be sufficient for a turn of the SAS-6 spiral to form (4.5 SAS-6 dimers/turn), but not for a complete SAS-6 ring (9 SAS-6 dimers/ring). In the revised discussion we present this line of thought more clearly, but also stress that it is simply a hypothesis.

*You propose that SAS-5 multimerization is critical for organizing SAS-6 complexes into spirals with 9-fold symmetry (see the last sentence of the Abstract, the last sentence of the Introduction, and the Discussion). Since this hypothesis has not been tested, I feel that this statement is inappropriate for the Abstract and Introduction. More work could be done to see if purified SAS-5 affects SAS-6 assembly* in vitro*, but I think that is beyond the scope of this paper. However, I think speculation is appropriate for the Discussion section. Furthermore, in the Discussion section the authors should also consider the possibility that the predicted Beta sheet spanning a.a. 282–295 in SAS-5 might be physiologically relevant and promote further multimerization beyond hexamers. The authors cannot dismiss this possibility since they never tested the functionality of SAS-5*_*Δ282-295*_ in vivo.

We have removed references to SAS-5 assisting SAS-6 organisation into spirals with 9-fold symmetry from the Abstract, and make it clear at the end of the Introduction that the SAS-5 oligomerisation driving SAS-6 self-assembly is merely a hypothesis, which is now expanded on only in the Discussion section. Also in the Discussion, we now acknowledge that SAS-5 polymerisation through the predicted beta-strand may be physiologically relevant, although we do not expect this to be the case as the aggregates of SAS-5 tend to be as big as centrioles. Should these SAS-5 aggregates occur in vivo, they would leave little space for SAS-6, SAS-4 and microtubules.

*The model in*
Figure 8
*is misleading since the authors never tested if SAS-5 oligomerization affects SAS-6 organization. This figure should be removed*.

The reviewers are correct that our work does not directly test the effect of SAS-5 on SAS-6 organisation. However, we believe that Figure 8 is valuable as it presents a hypothesis (or working model) for SAS-5 function that is consistent with existing results. Our work clearly shows that SAS-5 oligomerises and that this oligomerisation is necessary for function. Further, it has long been known (e.g. [42]) that SAS-5 function is needed for SAS-6 self-assembly in *C. elegans* centrioles. Figure 8 brings these two ideas together in a schematic form.

We believe that models of this type make manuscripts approachable to a wider audience, thus we have decided to retain Figure 8 in the revised manuscript. Nevertheless, we have modified the Discussion and Figure 8 legend to make clear that this model is simply a hypothesis. However, should you (similar to the reviewers) feel strongly about removing Figure 8, we would be prepared to move it into a supplementary panel related to the new Figure 7.

*No reference is made to an earlier characterization of SAS-5 structure published by*
[65]
*(Worm). The authors should mention this earlier work, why their results are different, and remove the lines in the Discussion section that claim that their study is the first structural and functional analysis of SAS-5 architecture*.

Shimanovskaya et al. suggested that SAS-5 forms tetramers in solution based on SEC-MALS and SAXS data, and that the SAS-5 N-terminal region ahead of the coiled coil of this protein is unstructured. However, the authors of this study did not provide a structural basis for SAS-5 oligomerisation, nor did they provide a combined SAS-5 structure-function relationship. We now reference this work in appropriate sections of our revised manuscript (in the Results), and we removed the statement of priority. Comparison of the SAS-5 oligomeric state suggested in Shimanovskaya et al. with our work is, however, difficult. The authors of the earlier paper only show SEC-MALS data from single, unspecified, concentrations of their different SAS-5 constructs. The raw SAXS data are not shown, nor is there an analysis of SAXS data as a function of SAS-5 concentration.

*The MALS experiments with the coiled coil construct have not been done at higher concentrations, the use of MD analysis to demonstrate trimerisation of the coiled coil is rather weak evidence and some of the MALS runs in*
Figure 2
*are a bit puzzling: The broad black curve gives a flat MW throughout the peak and the tail, while the brown curves that elute at that tail position are less than half that MW. If the assembly goes concentration dependent monomer-dimer-trimer then that behaviour is a bit hard to understand as the black curve should reflect these steps throughout its very broad peak*.

We are puzzled by these comments. SEC-MALS experiments of SAS-5 coiled coils (for all nematode species) have been done up to on-column concentrations of ∼150 μM or more (see Figure 2, Figure 2—figure supplement 2 and Table 2), which required the injection of protein samples with concentration in excess of 4 mM. In both Figure 2 and Figure 2—figure supplement 2, the SAS-5 coiled coils can be clearly seen to be capped at the trimer size over the last two / three data series (typically from an on-column concentration of 60 μM and higher). We believe this is a strong argument for the trimeric nature of this coiled coil; the MD analysis was performed to “undo” the crystal packing distortions of the coiled coil, rather than to prove its trimerisation. Finally, the broad black peaks of Figure 2 (and similar ones in Figure 2—figure supplement 2) are the result of overloading the size exclusion column with protein in our last one or two data series. In these cases the broadness of the SEC-MALS peak does not represent protein equilibria, as the reviewers suggest, but simply a protein overload. For example, the protein on-column concentration over the flat MW section of the broad peak in Figure 2 is always in excess of 30 μM, as calculated from the differential refractive index. We have added a note in figure legends to explain the origin of these broad peaks as protein overloads.

*It is not always 100% clear whether the solubility tag has been removed or left on*.

We have modified the text and figure legends to explicitly state when the solubility tag was present in the proteins utilised.

*CD deconvolution into secondary structure percentages tends not to be very accurate in terms of exact numbers and I would consider just mentioning that constructs where found to be mainly helical and so on*.

The reviewers are correct to suggest that CD deconvolution is not accurate enough to provide absolute percentage terms of secondary structure, which is why in the text we never refer to the constructs in terms other than “mainly helical” etc. However, we were very careful in doing all CD measurements and data analysis under closely matched conditions, which is why we believe that the relative secondary structure changes seen in CD deconvolution analysis have merit. This is especially valuable when comparing wild-type and mutant proteins, as in Figure 5—figure supplement 1, where CD deconvolution can assist readers with less expertise in interpreting the raw CD data. Thus, we would prefer not to remove the CD deconvolution bars from the figures, but we rather explicitly state in figure legends that this type of analysis is semi-quantitative.

*I'm surprised that the authors didn't test if SAS-5 multimerization affects SAS-6 recruitment* in vivo*. The strong centriole/cell division phenotype seen in the L141E and I247E mutants might be due to the fact that not enough SAS-6 is recruited to the nascent daughter centriole, rather than SAS-6 not being organized properly once delivered there. The authors could use immunofluorescence to test this possibility*.

This is an interesting suggestion. Unfortunately, however, an immunofluorescence experiment using antibodies directed against SAS-6 cannot ascertain whether or not SAS-6 is recruited in the L141E and I247E mutant conditions. Indeed, SAS-6 at centrioles inherited from sperm is extremely stable, as recently reported (PMID 25906994). Moreover, due to the minute size of centrioles in *C. elegans*, the nascent centriole in one-cell stage embryos cannot be resolved by confocal microscopy from the paternally contributed centrioles. Thus, two SAS-6 foci are observed in both wild type and *sas-5(RNAi)* one-cell stage embryos. Therefore, discriminating between a scenario in which the protein is not recruited at all from one in which it is recruited initially but not organized properly thereafter in the mutant conditions is not possible by this means. An alternative that could be envisaged would be to use a transgenic line expressing a detectable fusion protein between SAS-6 and a fluorescent protein other than GFP (e.g. mCherry-SAS-6), but this is not available. In the absence of such experiments, we now indicate the possibility mentioned by the reviewers in the revised manuscript (paragraph two, Discussion section).

*The model in*
Figure 4
*should be revised so that it is clear that the proposed model is for a modified version of SAS-5 that lacks the central predicted Beta sheet (SAS-5*_*FLEX*_*)*.

The former Figure 4 (now Figure 7) does not include the section where the predicted beta-strand would locate, but only the structured coiled coil and Implico segments. We now make it explicit in the figure legend that this oligomerisation model is valid for SAS-5 in the absence of the predicted beta-strand.